# Motion Matters: Compact Gaussian Streaming for Free-Viewpoint Video Reconstruction

Jiacong Chen[1,2]    Qingyu Mao[3]    Youneng Bao[4]    Xiandong Meng[5]    Fanyang Meng[5]
Ronggang Wang[5,6]    Yongsheng Liang[1,2✉]

[1]College of Applied Technology, Shenzhen University, Shenzhen, China
[2]College of Big Data and Internet, Shenzhen Technology University, Shenzhen, China
[3]College of Electronics and Information Engineering, Shenzhen University, Shenzhen, China
[4]Department of Computer Science, City University of Hong Kong, Hong Kong, China
[5]Pengcheng Laboratory, Shenzhen, China
[6]School of Electronic and Computer Engineering, Peking University, Shenzhen, China

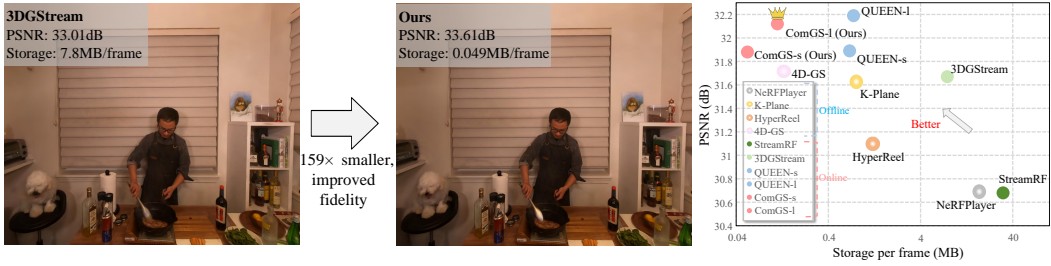

Figure 1: **Left**: Experimental results on N3DV dataset [1] showcase the effectiveness of our method, which reduces the storage requirement of 3DGStream [2] by 159 ×, with enhanced visual quality. **Right**: Comparison with existing methods in storage and reconstruction fidelity. Hollow circles denote offline methods, while solid circles represent online methods.

## Abstract

3D Gaussian Splatting (3DGS) has emerged as a high-fidelity and efficient paradigm for online free-viewpoint video (FVV) reconstruction, offering viewers rapid responsiveness and immersive experiences. However, existing online methods face challenge in prohibitive storage requirements primarily due to point-wise modeling that fails to exploit the motion properties. To address this limitation, we propose a novel Compact Gaussian Streaming (ComGS) framework, leveraging the locality and consistency of motion in dynamic scene, that models object-consistent Gaussian point motion through keypoint-driven motion representation. By transmitting only the keypoint attributes, this framework provides a more storage-efficient solution. Specifically, we first identify a sparse set of motion-sensitive keypoints localized within motion regions using a viewspace gradient difference strategy. Equipped with these keypoints, we propose an adaptive motion-driven mechanism that predicts a spatial influence field for propagating keypoint motion to neighboring Gaussian points with similar motion. Moreover, ComGS adopts an error-aware correction strategy for key frame reconstruction that selectively refines erroneous regions and mitigates error accumulation without unnecessary overhead. Overall, ComGS achieves a remarkable storage reduction of over 159 × compared to 3DGStream and 14 × compared to the SOTA method QUEEN, while maintaining competitive visual fidelity and rendering speed. Project page: *https://chenjiacong-1005.github.io/ComGS/*.

39th Conference on Neural Information Processing Systems (NeurIPS 2025).

# 1 Introduction

Reconstructing free-viewpoint video (FVV) from multi-view videos captured by cameras with known poses has attracted growing interest in the field of computer vision and graphics. FVV exhibits great potential as a next-generation visual medium that enables immersive and interactive experiences, with broad application in virtual and augmented reality (VR/AR) applications [3].

Recently, 3D Gaussian Splatting (3DGS) has become a promising method for FVV reconstruction, due to its significant advancements in real-time rendering and high-fidelity view synthesis. These approaches typically fall into two categories: 1) incorporating temporal function into Gaussian primitives and optimizing directly [4–6], and 2) applying a deformation field to capture the spatio-temporal transformations of canonical Gaussians [7–11]. While these FVV reconstructions accurately represent dynamic scenes, they are trained in an offline manner and require transmitting the full set of reconstructed parameters prior to rendering.

In contrast, by enabling per-frame training and progressive transmission, online FVV reconstruction allows immediate playback without the overhead of full-scene preloading. As a pioneer work, 3DGStream [2] extends 3DGS to online FVV reconstruction using InstantNGP [12] to model the geometric transformation frame-by-frame. While achieving impressive rendering speed, its structural constraint hinders the volumetric representation performance and degrades the visual quality. Building on this paradigm, subsequent works [13, 14] enhance model expressiveness through explicitly optimizing Gaussian attribute residuals, achieving competitive synthesis quality and higher robustness. However, the storage demands of these methods remain prohibitively high for real-time transmission, with reconstructed data typically exceeding 20 MB per second.

In this paper, we aim to design a storage-efficient solution for FVV streaming that minimizes bandwidth requirements and enables real-time transmission. In online FVV reconstruction, since dynamic scenes contain a large proportion of static regions, the key to efficient reconstruction lies in motion modeling. Our first insight, therefore, is to only model the Gaussian attribute residuals in the motion regions, which eliminates the unnecessary updates in static regions. Building on motion modeling, we note that scene motion tends to be consistent, where Gaussian points associated with the same object typically exhibit the same or similar motion in dynamic scene representation. Our second insight, based on this observation, is to use a shared motion representation to model the attribute residuals with similar motion. This contrasts with existing online methods [2, 13] that utilize point-wise strategy to update the attribute residuals in motion regions, and the result is motion redundancy elimination and more compact storage. Lastly, we exploit a key frame fine-tune strategy to handle the error accumulation brought by non-rigid motion and novel objects emergence.

Specifically, to accomplish this, we propose a Compact Gaussian Streaming (ComGS) framework that leverages a set of keypoints ($= 200$), significantly fewer than the full set of Gaussian points ($\approx 200K$), to holistically model motion regions at each timestep. ComGS begins with a motion-sensitive keypoints selection through a viewspace gradient difference strategy. This ensures that the selected keypoints are accurately positioned within motion regions and prevents redundant or incorrect modeling of static areas. Subsequently, we design an adaptive motion-driven mechanism that defines a keypoint-specific spatial influence field, with which neighboring Gaussian points can share the motion of the keypoint. Unlike conventional k-nearest neighbor (KNN) methods [15, 16], the spatial influence field can accommodate the complexity and variability of motion structure in dynamic scenes, so that keypoints can more accurately drive the motion of the surrounding region. Finally, to mitigate error accumulation in a compact and effective manner, we propose an error-aware correction strategy for key frame reconstruction that selectively updates only those Gaussians with reconstruction errors.

Our major contributions can be summarized as follow:

- We introduce a motion-sensitive keypoint selection to accurately identify keypoints within motion regions, and an adaptive motion-driven mechanism that effectively propagates motion to neighboring points. These leverage the locality and consistency of motion and achieve a more storage-efficient solution for online FVV reconstruction.

- We propose an error-aware correction strategy for key frame reconstruction that mitigates error accumulation over time by selectively updating Gaussian points with reconstruction errors, which ensures long-term consistency and minimizes redundant correction.

- Experiments on two benchmark datasets show that the effectiveness of our method and its individual components. Our method achieves a compression ratio of $159\times$ over the 3DGStream and $14\times$ over state-of-the-art model QUEEN, enabling real-time transmission while preserving competitive reconstruction quality and rendering speed.

## 2 Related work

### 2.1 Dynamic Gaussian Splatting

Recently, 3D Gaussian Splatting (3DGS) [17–22] has attracted great attention in Free-viewpoint video (FVV) reconstruction for its high photorealistic performance and real-time rendering speed. Several works [4–6, 23, 24] expand temporal variation as a function and optimize directly for modeling Gaussian attributes across frames. For instance, 4D Gaussian Splatting [4] incorporates time-conditioned 3D Gaussians and auxiliary components into 4D Gaussians, while ST-GS [6] models the transformation of structural attributes and opacity as a temporal function to represent scene motions. To support long FVVs representation, TGH [23] introduces a multi-level hierarchy of 4D Gaussian primitives that exploits various degrees of temporal redundancy in dynamic scenes. While these time variant-based methods achieve superior rendering efficiency, they often suffer from prohibitive storage requirements. Other works [8, 11, 25–27] employ vanilla 3D Gaussians as a canonical space and a deformation field to represent the dynamic scene. In this category, 4D-GS [8] utilizes hexplanes [28], six orthogonal planes, as latent embeddings and deliver them into a small MLP to deform temporal transformation of Gaussian points, achieving efficient computational complexity and lightweight storage requirement. Building upon this, GD-GS [11] further improves scene modeling accuracy by incorporating geometric priors, which provides a more structured and precise representation of dynamic scene. Among them, both SC-GS [15] and SP-GS [16] adopt sparse control points to control scene motion using a k-nearest neighbor (KNN) [29] strategy for motion modeling. While these methods achieve notable improvements in computational efficiency and rendering speed, they are designed for offline FVV reconstruction and do not support frame-by-frame delivering. Additionally, motion-insensitive control point selection and scale-agnostic KNN motion modeling lead to redundant representation of static regions and reduced deformation accuracy in dynamic scenes. Our online method addresses these limitations by selecting keypoints from motion regions at each timestep and modeling motion with awareness of local motion scales, which enables more accurate and efficient modeling of online FVV.

### 2.2 Online Free-Viewpoint Video Reconstruction

Compared to the offline methods, online reconstruction enables FVV to be incrementally trained and transmitted in a per-frame manner, which allows users to preview or interact immediately with the video content. Leveraging the high-fidelity view synthesis capabilities of Neural Radiance Field (NeRF) [12, 30–36], a set of studies have explored NeRF-based methods [37–41] for online FVV reconstruction, such as StreamRF [37], VideoRF [38] and TeTriRF [40]. Despite advanced visual quality, NeRF-based methods are hindered by their limited rendering speeding of implicit structure, which limits their practical applications.

With the utilization of 3DGS [17], 3DGStream [2] introduces a hash-based MLP to encode the position and rotation transformation of Gaussian points at each frame, and designs an adaptive Gaussians addition strategy for novel objects across frames. Based on this paradigm, QUEEN [13] proposes a Gaussian residual-based framework for model expressiveness enhancement and a learned quantization-sparsity framework for residuals compression. HiCoM [14] designs a hierarchical coherent motion mechanism to effectively capture and represent scene motion for fast and accurate training. To deploy into mobile device, $V^3$ [42] presents a novel approach that compresses Gaussian attributes as a 2D video to facilitate hardware video codecs. IGS [43] proposes a generalized anchor-driven Gaussian motion network that learns residuals with a singe step, achieving a significant improvement of training speed. Nevertheless, these methods face challenge in real-time transmission, due to their substantial storage requirements. This overhead mainly stems from redundant updates of static Gaussian points across frames, as well as repeated modeling of Gaussian points with similar motion. Our study exploits the locality and consistency of motion by leveraging motion-sensitive keypoints to adaptively drive motion regions, and this avoids redundant storage and transmission.

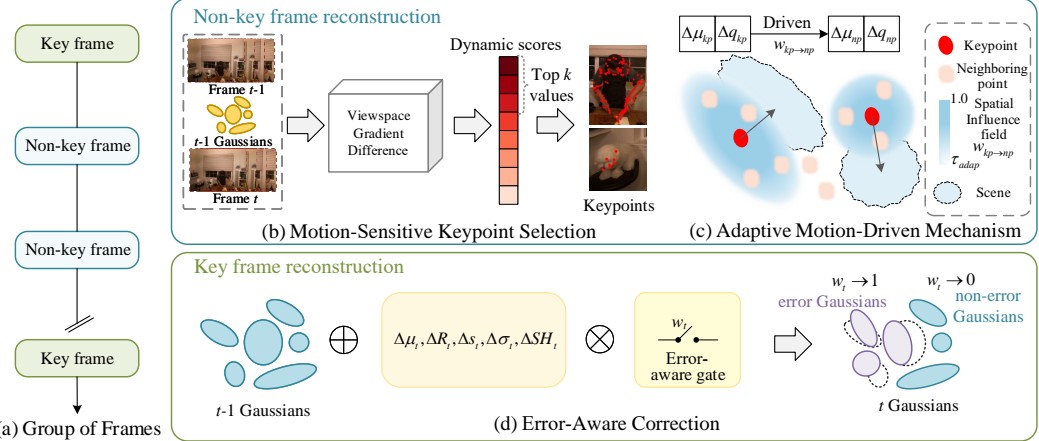

(a) Group of Frames

Figure 2: The overall pipeline of ComGS framework. (a) The reconstruction process starts from the first frame initialized using vanilla 3DGS [17]. Subsequent frames are organized into groups of frames (GoFs). For non-key frames, (b) we begins with a motion-sensitive keypoint selection using a viewspace gradient difference strategy, (c) and utilizes an adaptive motion-driven mechanism to control neighboring points motion. For key frames, (d) an error-aware correction strategy is introduced to mitigate the error accumulation across frames.

## 2.3 3D Gaussian Representation Compression

Despite 3DGS-based methods achieve impressive performance in novel view synthesis [17, 44], the massive size of Gaussian points hinder them for efficient storage and transmission. Several studies propose a variety of compression techniques for reducing the required storage, which can be categorized into either post-processing-based [19–21, 45, 13] or neural contextual coding-based methods [10, 46–49]. Post-processing-based approaches include removing unimportant Gaussian points [19, 20], pruning spherical harmonic coefficients [19, 21], and applying vector quantization [13, 45] to compress the parameter representation. The latter methods utilize sophisticated entropy modeling to accurately predict probability distributions that exploit global context for compressing 3D Gaussian representation more effectively. In this paper, we focus on leveraging the locality and consistency of motion in dynamic scene and mitigating the redundancy reconstruction on static and similar motion regions, introducing a novel and more compact method for online FVV reconstruction.

## 3 Methods

Our goal is to reconstruct and transmit FVV in a storage-efficient and streaming manner. To achieve it, we propose a Compact Gaussian Streaming (ComGS) framework for online FVV reconstruction, as illustrated in Fig. 2. First, ComGS begins with a motion-sensitivity keypoint selection using a viewspace gradient difference, ensuring subsequent motion control learning (Sec. 3.2). Second, we develop an adaptive motion-driven mechanism that applies a spatial influence field to control neighboring point motion (Sec. 3.3). Third, we devise an error-aware correction strategy for key frame reconstruction to mitigate error accumulation brought by non-rigid motion and novel objects emergence in online reconstruction (Sec. 3.4). Finally, we introduce our compression techniques and optimization process in Sec. 3.5.

## 3.1 Preliminary

3DGS [17] models a 3D scene as a large amount of anisotropic 3D Gaussian points in world space as an explicit representation. The central position and geometric shape of each Gaussian point $i$ in world space are defined by a mean vector $\mu_i$ and covariance matrix $\Sigma_i$, mathematically represented as:

$$G_i(x) = \exp(-\frac{1}{2}(\mathbf{x} - \mu_i)^T \Sigma_i^{-1} (\mathbf{x} - \mu_i)) \tag{1}$$

For differentiable optimization, the covariance matrix $\Sigma_i$ is decoupled into a scaling matrix $S_i$ and a rotation matrix $R_i$. Each Gaussian point is characterized by its color $c_i$ and opacity $\sigma_i$. For novel view synthesis, the covariance matrix $\Sigma_i'$ in camera coordinate is given as:

$$\Sigma_i' = \mathbf{J}\mathbf{W}\Sigma_i\mathbf{W}^T\mathbf{J}^T \tag{2}$$

where $\mathbf{J}$ is the Jacobian of the affine approximation of the perspective projection and $\mathbf{W}$ represents the view transformation matrix mapping world coordinates.

During rendering, the Gaussian points are initially projected into viewing plane, and the final color $C$ can be obtained by $\alpha$-blending of the $N$ ordered 3D Gaussian points overlapping the pixel as:

$$C = \sum_{i=1}^{N} c_i\alpha_i \prod_{j=1}^{i-1}(1 - \alpha_j) \tag{3}$$

where $\alpha_i$ represents the blending weight of the $i^{th}$ Gaussian point.

### 3.2 Motion-Sensitive Keypoint Selection

Establishing an effective keypoint-driven motion representation necessitates to select appropriate keypoints. Considering motion locality, keypoints should be located in motion regions, which avoids redundant modeling in static areas and enables accurate modeling of complex motions

Thus, inspired by [13], we propose a motion-sensitive keypoint selection based on viewspace gradient difference (Fig. 2 (b)). The core idea is to identify the dynamic Gaussian by the gradient change of rendering loss in inter-frames, and based on the gradient values, the $k$ Gaussian points with the largest gradient are selected as keypoints. Specifically, following the gradient computation in 3DGS [17], we compute gradients using the previous Gaussian positions $p_{t-1}$, the rendered images $\hat{I}_{t-1}$, the reconstruction loss $\mathcal{L}_{recon}$, and the ground-truth images $I_{t-1}$ and $I_t$:

$$\mathcal{G}_{t-1} = \frac{\partial \mathcal{L}_{recon}^{t-1}}{\partial p_{t-1}}, \quad \mathcal{L}_{recon}^{t-1} = \mathcal{L}_{recon}(\hat{I}_{t-1}, I_{t-1}) \tag{4}$$

$$\mathcal{G}_t = \frac{\partial \mathcal{L}_{recon}^{t}}{\partial p_{t-1}}, \quad \mathcal{L}_{recon}^{t} = \mathcal{L}_{recon}(\hat{I}_{t-1}, I_t) \tag{5}$$

Dynamic significance scores $\Delta\mathcal{G}_t \in \mathbb{R}^N$ ($N$ is the number of Gaussians) were calculated by means of absolute values of gradient differences:

$$\Delta\mathcal{G}_t = \frac{1}{V}\sum_{v=1}^{V}|\mathcal{G}_t^{(v)} - \mathcal{G}_{t-1}^{(v)}| \tag{6}$$

where $V$ is the number of the training viewpoints. Finally, we select the top $k$ high dynamic significance scores from all Gaussian points as keypoints $\mathcal{K}_t$ at timestamp $t$. Selecting the top-$k$ Gaussian points with the highest dynamic scores not only identifies those located in motion regions, but also naturally allocates more keypoints to the areas with complex motion, facilitating more accurate modeling of such regions.

In this paper, for a balance of training efficiency and reconstructed quality, we set $k = 200$.

### 3.3 Adaptive Motion-Driven Mechanism

Equipped with the selected keypoints $\mathcal{K}_t$ at current timestep, the next step is to determine which neighboring points are controlled by these keypoints, and apply their transformations to drive the motion of the controlled neighboring points. Previous works [15, 16] employ k-nearest neighbor (KNN) [29] search to predict the motion of each Gaussian points, showing advanced results in monocular synthetic video reconstruction, but they do not fully consider unnecessary modeling in static region and motion scale difference, which leads to computational redundancy and inaccurate representation.

In contrast, we propose an adaptive motion-driven mechanism that enables each keypoint to drive neighboring points through a spatial influence field, as illustrated Fig. 2 (c). Specifically, motivated

by [17], for each keypoints $\mathcal{K}_t^i$ at $t$ timestep, we initialize a quaternion $q_{adap}^i \in \mathbb{R}^4$ and a scaling vector $s_{adap}^i \in \mathbb{R}^3$ to compute the spatial influence field $\Sigma_{adap}^i \in \mathbb{R}^{3\times3}$. For a neighboring Gaussian point $G_j$ with position $\mu_j$, its distance to keypoint $\mathcal{K}_t^i$ is given by $d_{ij} = \mu_j - \mu_{\mathcal{K}_t^i}$. The influence weight is then computed as:

$$w_{ij} = \exp\left(-\frac{1}{2}d_{ij}^\top(\Sigma_{adap}^i)^{-1}d_{ij}\right) \tag{7}$$

If $w_{ij}$ exceeds a predefined threshold $\tau_{adap}$, the Gaussian $G_j$ is considered to be controlled by keypoint $\mathcal{K}_t^i$:

$$\mathcal{C}_t^i = \{G_j \mid w_{ij} \geq \tau_{adap}\} \tag{8}$$

where $\mathcal{C}_t^i$ denotes the set of Gaussian points controlled by keypoint $\mathcal{K}_t^i$.

To model motion, each keypoint $\mathcal{K}_t^i$ is further assigned a learnable translation offset $\Delta\mu_{\mathcal{K}_t^i} \in \mathbb{R}^3$ and a rotation represented by a quaternion $\Delta q_{\mathcal{K}_t^i} \in \mathbb{R}^4$. For a Gaussian $G_j$ controlled by multiple keypoints $\{\mathcal{K}_t^i\}_{i\in\mathcal{I}_t^j}$, its overall motion is computed by aggregating the motions of its associated keypoints, weighted by their influence scores $w_{ij}$:

$$\Delta\mu_t^j = \sum_{i\in\mathcal{I}_t^j} w_{ij} \cdot \Delta\mu_{\mathcal{K}_t^i}, \quad \Delta q_t^j = \sum_{i\in\mathcal{I}_t^j} w_{ij} \cdot \Delta q_{\mathcal{K}_t^i} \tag{9}$$

where $\Delta\mu_t^j$ and $\Delta q_t^j$ indicate the transformation of Gaussian $j$ at $t$ timestep, and $\mathcal{I}_t^j$ represents the set of keypoints that control the motion of Gaussian $j$.

By leveraging a compact set of keypoints with spatial influence fields, our method enables accurate and efficient control of Gaussian motions at each frame. Since Gaussians share motion attributes through keypoints, only 14 parameters per keypoint are required, significantly reducing storage demands and mitigating data redundancy.

### 3.4 Error-Aware Corrector

By using keypoints to drive scene motion, we model the transformation of Gaussian points from the previous frame to the current frame with an extremely compact parameters. Nevertheless, keypoint-based motion controlling only supports to represent rigid motion effectively and faces challenge to handle non-rigid motion and novel objects emergence, which results in error accumulation across frames.

A straightforward solution to mitigate error accumulation and ensure accurate long-term FVV representations is to separate the video into frame groups and update the attributes of all Gaussian points at key frames. However, this strategy would lead to a substantial of unnecessary parameters updating, since most of parameters are already correctly representing the scene and do not require modification. To mitigate error accumulation in a compact and efficient manner, we propose an error-aware corrector strategy that only finetunes the Gaussian points with detected errors, significantly decreasing storage demands and promoting more accurate scene reconstruction, as illustrated in Fig. 2 (d).

Specifically, given a video sequence, we select a key frame every $s$ frames, forming the key frame sequence $\{f_s, f_{2s}, \ldots, f_{ns}\}$, as shown in Fig. 2 (a). Note that in this paper, key frames are used for error correction, and only the first frame of the video sequence is independently reconstructed. The remaining frames are reconstructed by keypoints driven. During key frame reconstruction, given the attributes of a Gaussian point at previous timestep $\theta_i^{t-1} : (\mu_i^{t-1}, q_i^{t-1}, s_i^{t-1}, \sigma_i^{t-1}, c_i^{t-1})$, we introduce a set of learnable parameters $\Delta\theta_i^t$ to model the attribute residuals. To identify which Gaussian points require correction, we predict a learnable mask $m_i$ for each point. A sigmoid function is used to map $m_i$ to the range $(0, 1)$, which refers as a soft mask:

$$m_i^{soft} = Sigmoid(m_i), m_i^{soft} \in (0, 1) \tag{10}$$

Similar to [20, 21], the soft mask is binarized into a hard mask using a predefined threshold $\phi_{thres}$, where the non-differentiable binarization is handled with the straight-through estimator (STE) to enable gradient flow, represented as:

$$m_i^{hard} = sg(\mathbb{1}(m_i^{soft} > \phi_{thres}) - m_i^{soft}) + m_i^{soft}, m_i^{hard} \in \{0, 1\} \tag{11}$$

Table 1: Quantitative comparisons on Neural 3D Video (N3DV) [1] and MeetRoom datasets [37].

| Dataset | Category | Method | PSNR (dB) ↑ | SSIM ↑ | LPIPS ↓ | Storage (MB) ↓ | Training (sec) ↓ | Rendering (FPS) ↑ |
|---------|----------|--------|-------------|--------|---------|----------------|------------------|-------------------|
| N3DV | Offline | NeRFPlayer [41] | 30.69 | 0.932 | 0.209 | 17.10 | 72 | 0.05 |
| | | HyperReel [52] | 31.10 | 0.928 | - | 1.20 | 104 | 2.00 |
| | | 4D-GS [8] | 31.15 | 0.964 | 0.149 | 0.13 | 8 | 34 |
| | | SpaceTime [6] | 32.05 | 0.948 | - | 0.67 | 20 | 140 |
| | Online | StreamRF [37] | 30.68 | - | - | 31.4 | 15 | 8.3 |
| | | TeTriRF [40] | 30.43 | 0.906 | 0.248 | 0.06 | 39 | 4 |
| | | 3DGStream [2] | 31.67 | 0.941 | 0.140 | 7.80 | 8.5 | 261 |
| | | QUEEN-s [13] | 31.89 | 0.945 | 0.139 | 0.68 | 4.65 | 345 |
| | | QUEEN-l [13] | 32.19 | 0.946 | 0.136 | 0.75 | 7.9 | 248 |
| | | ComGS-s (ours) | 31.87 | 0.943 | 0.132 | 0.049 | 37 | 91 |
| | | ComGS-l (ours) | 32.12 | 0.945 | 0.129 | 0.106 | 43 | 147 |
| MeetRoom | Static | I-NGP [53] | 28.10 | - | - | 48.2 | 66 | 4.1 |
| | | 3DG-S [17] | 31.31 | - | - | 21.1 | 156 | 571 |
| | Online | StreamRF [37] | 26.72 | - | - | 9.0 | 10.2 | 10 |
| | | 3DGStream [2] | 30.79 | 0.950 | 0.188 | 4.1 | 4.9 | 350 |
| | | QUEEN-s [13] | 31.14 | 0.954 | 0.173 | 0.45 | 3.8 | 421 |
| | | ComGS-s (ours) | 31.49 | 0.955 | 0.171 | 0.028 | 28.3 | 98 |

where $\mathbb{1}$ is the indicator function and $sg$ indicates the stop gradient operation. Then, the $m_i^{hard}$ is applied to the attribute residuals before rendering, followed as:

$$\theta_i^t = \theta_i^{t-1} + m_i^{hard}\Delta\theta_i^t \tag{12}$$

Meanwhile, we define a optimized function to regulate the perceptual error while encouraging sparse residual updates:

$$\mathcal{L}_{error} = \frac{1}{N}\sum_i m_i^{soft} \tag{13}$$

where $N$ is the number of all Gaussian points. After optimization for the current key frame, only the attribute residuals $\hat{\Delta\theta}^t = \{\Delta\theta_i^t | m_i^{hard} = 1\}$ and the hard mask set $\mathcal{M}^{hard} = \{m_i^{hard} | i = 1, 2, ..., N\}$ need to be stored and transmitted, minimizing the required data redundancy and transmission overhead.

### 3.5 Optimization and Compression

For the first frame optimization, we employ COLMAP [50] to generate the initial point cloud and follow the pipeline of 3DGStream [2]. The optimization for both the first frame and non-key frames is supervised by the reconstruction loss $\mathcal{L}_{recon}$, which is composed by an $L_1$-norm loss $\mathcal{L}_1$ and a D-SSIM loss $\mathcal{L}_{D-SSIM}$ [51]:

$$\mathcal{L}_{recon} = (1 - \lambda_{D-SSIM})\mathcal{L}_1 + \lambda_{D-SSIM}\mathcal{L}_{D-SSIM} \tag{14}$$

For key frame optimization, we minimize a combined loss consisting of $\mathcal{L}_{recon}$ and $\mathcal{L}_{error}$:

$$\mathcal{L}_{total} = \mathcal{L}_{recon} + \lambda_{error}\mathcal{L}_{error} \tag{15}$$

where $\lambda_{error}$ controls the degree of error awareness, thereby balancing reconstruction quality and memory efficiency. We set $\lambda_{D-SSIM} = 0.2$ and $\lambda_{error} = 0.001$ in this paper.

After optimization, the initialized Gaussians $\theta^0$ and the residuals $\hat{\Delta\theta}^t$ for key frame error correction are further compressed through quantization and entropy coding, enabling compact storage without performance degradation. More details are provided in the **Appendix**.

## 4 Experiments

### 4.1 Experimental Setup

We evaluate our method on two widely-used public benchmark datasets. **(1) Neural 3D Video (N3DV)** [1] consists of six indoor video sequences captured by 18 to 21 viewpoints. **(2) Meet**

Table 2: Quantitative comparisons on the long video sequence *flame salmon* from the N3DV dataset [1].

| Method | PSNR (dB) ↑ | SSIM ↑ | LPIPS ↓ | Storage (MB) ↓ | Training (sec) ↓ | Rendering (FPS) ↑ |
|---|---|---|---|---|---|---|
| E-NeRF [54] | 23.48 | 0.89 | 0.260 | 0.692 | 13.8 | 5 |
| 4DGS [4] | 28.89 | 0.952 | 0.197 | 2.23 | 31.2 | 90 |
| TGH [23] | 29.44 | 0.945 | 0.214 | 0.075 | 6.3 | 550 |
| ComGS-s (ours) | 29.56 | 0.920 | 0.140 | 0.053 | 45.4 | 91 |

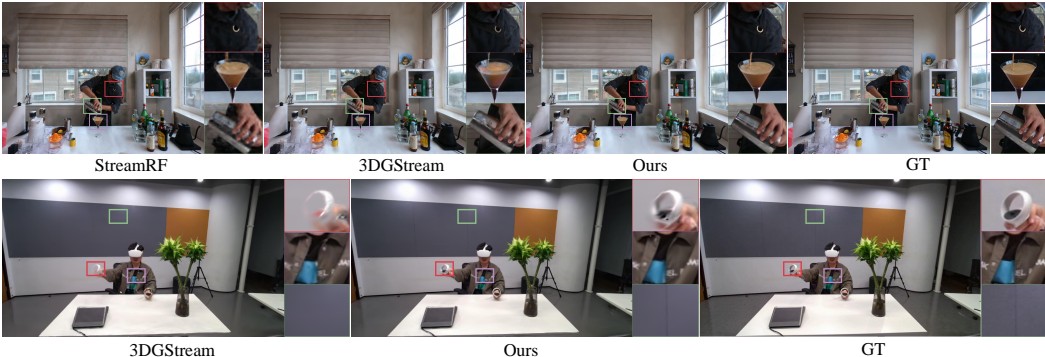

Figure 3: Quantitative comparison. We visualize our method and other online FVV methods on N3DV [1] and MeetRoom [37] dataset.

**Room** [37] comprises four indoor scenes recorded with a 13 cameras multi-view system. In both of two datasets, we employ the first view for testing. Our method is implemented on an NVIDIA A100 GPU. We train 150 epochs for non-key frames reconstruction and 1000 epochs for key frames fine-tuning. We measure the visual quality of rendered images by average PSNR, required storage, rendering FPS and training time. More implement details are provided in the **Appendix**.

### 4.2 Quantitative Comparisons

We conduct quantitative comparisons on existing online methods including StreamRF [37], TeTriRF [40], 3DGStream [2] and QUEEN [13], as well as the SOTA offline FVV approaches [6, 8, 41, 52] on N3DV and Meetroom (Tab. 1). Our method is evaluated in two variants: ComGS-s (small) and ComGS-l (large), using key frame intervals of $s=10$ and $s=2$, respectively.

Tab. 1 shows that our ComGS achieves competitive results among existing online FVV methods on N3DV dataset. Notably, ComGS-s achieves a substantial reduction in storage by $159\times$ compared to 3DGStream and $14\times$ compared to QUEEN. This advantage enables real-time transmission in limited bandwidth and enhances the overall user viewing experience. On MeetRoom dataset, our method outperforms 3DGStream, obtaining +0.7dB PSNR and $146\times$ smaller size. Our advantages are mainly due to two factor: 1) using keypoint as a shared representation requires transmitting only a small number of keypoint attributes; and 2) the error-aware correction module effectively rectifies regions with scene inaccuracies using minimal additional parameters. In the **Appendix**, the quantitative results are provided for each scene to offer a more detailed comparison.

We further evaluate the effectiveness of ComGS on handling long videos. We compare our method with the TGH [23] (which is the most recently proposed for handling long video sequences) on the *Flame Salmon* sequence (1200 frames) from the N3DV dataset [1]. Tab. 2 shows that our method achieves competitive results on rendering quality and required storage.

### 4.3 Qualitative Comparisons

As shown in Fig. 3, we compare our reconstructed results to other online FVV methods on N3DV and MeetRoom. ComGS effectively reconstructs both motion and static regions and provides more closer results to the ground truth. Fig. 3 shows that 3DGStream introduces noticeable artifacts due to its global update of Gaussian points across the entire scene, which often leads to incorrect updates

Table 3: Ablation study on proposed components. *Flame Steak* and *Flame Salmon* scenes are from the N3DV dataset.

| Experiments | Selection | Adaptive | Correction | Flame Steak | | Flame Salmon | |
|---|---|---|---|---|---|---|---|
| | | | | PSNR (dB)↑ | Storage (KB)↓ | PSNR (dB)↑ | Storage (KB)↓ |
| 1 | × | ✓ | ✓ | 33.27 | 46.7 | 29.22 | 56.7 |
| 2 | ✓ | × | ✓ | 32.82 | 36.4 | 28.96 | 45.7 |
| 3 | × | × | ✓ | 31.26 | 37.9 | 27.75 | 46.4 |
| 4 | ✓ | ✓ | × | 31.67 | **26.9** | 28.74 | **26.9** |
| 5 | ✓ | ✓ | ✓ | **33.49** | 46.5 | **29.32** | 53.4 |

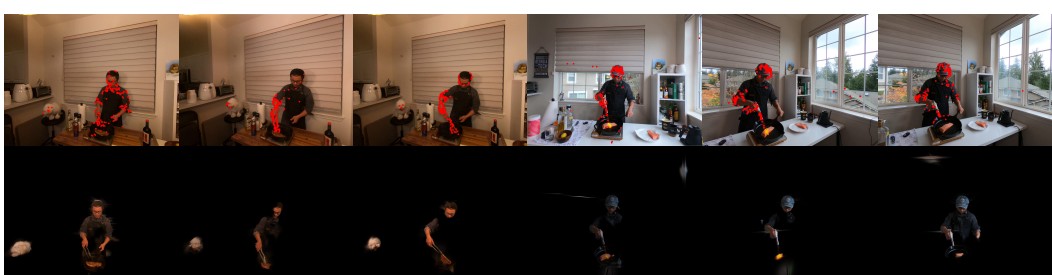

Figure 4: Visualization of our keypoint-driven motion representation. **Top**: selected keypoints are concentrated in motion regions. **Bottom**: adaptive control of neighboring points also focuses on motion-intensive areas, enabling accurate and efficient motion modeling.

in static regions. In contrast, our method restricts modeling to motion regions and applies targeted corrections in error-prone areas, resulting in more accurate and robust scene reconstruction. More qualitative results are offered in **Appendix**.

## 4.4 Ablation Study

To validate the effectiveness of our proposed methods, we ablate three components of ComGS framework in Tab. 3.

In the **Experiment 1**, we ablate the motion-sensitive keypoint selection and instead select keypoints randomly. Since the random selection is not guided by motion regions, it may result in ineffective modeling of static areas and inadequate representation on motion regions (Fig. 5 (b)), which leads to a slight degradation in PSNR. **Experiment 2** removes the adaptive motion-driven mechanism and models scene motion only using the selected keypoints, without incorporating any neighboring points. The resulting drop in reconstruction quality demonstrates that effective motion modeling relies not only on accurately keypoint selection, but also on the selection of their neighboring points. In the **Experiment 3**, we reconstruct FVV only relying on the first frame reconstruction and key frame correction, without modeling non-key frames by keypoint reconstruction, which results in a significant performance drop. We emphasize that although the parameters of keypoints are few, the keypoint-based modeling plays a crucial role in FVV reconstruction. **Experiment 4** ablates the error-aware correction in key frame reconstruction. The performance degradation demonstrates that the error-aware correction in key frames would solve the error accumulation across frames.

Table 4: Ablation study on comparing control strategies for neighboring points.

| Control tech | PSNR (dB) | Storage (KB) |
|---|---|---|
| KNN | 31.39 | **44.1** |
| Adaptive | **31.87** | 49.0 |

Table 5: Ablation study of the error-aware correction strategy.

| Configuration | PSNR (dB) | Storage (KB) |
|---|---|---|
| w/o error-aware | 31.65 | 373 |
| with error-aware | **31.87** | **49.0** |

To further investigate the role of keypoint-driven motion representation, we visualize the selection and driven process in Fig. 4. The top row shows that keypoints are predominantly selected in motion

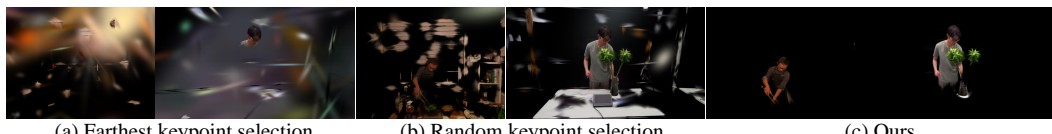

(a) Farthest keypoint selection  (b) Random keypoint selection  (c) Ours

Figure 5: Visualization of different selection methods and corresponding updated regions.

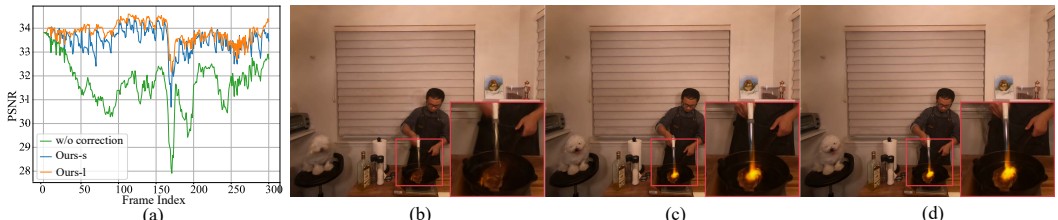

(a)    (b)    (c)    (d)

Figure 6: (a) PSNR comparison over time. Visualizations of (b) w/o key frames correction. (c) ComGS-s. (d) ComGS-l.

regions, such as the human body and moving objects. The bottom row highlights the adaptively controlled areas for neighboring points, which similarly focus on regions with significant motion (e.g., the person and the dog). Fig. 5 visualizes Gaussians updated region using farthest keypoint selection [16], random keypoint selection and our method, respectively, which demonstrates that our method accurately captures motion-intensive areas. These results indicate that ComGS can effectively leverage the locality and consistency of scene motion.

We also evaluate a KNN-based method [29] for selecting neighboring points around keypoints (Tab. 4). This approach shows inferior performance, as it does not distinguish between static and motion regions, leading to redundant modeling and poor adaptation to varying motion scales.

Fig. 6 evaluates the effect of key frame correction. The visual results in Fig. 6 (b–d) further highlight that key frame correction significantly reduces artifacts in motion regions such as flames, helping to maintain finer temporal consistency throughout the sequence. Tab. 5 shows that correction without error-aware leads to significantly higher storage due to redundant Gaussians updating. Moreover, without focusing on high-error regions, updates may affect error-free areas and result in suboptimal performance. Therefore, enabling error-awareness improves both accuracy and efficiency.

## 5  Conclusion

In this paper, we proposed ComGS, a storage-efficient framework for online FVV real-time transmission. We utilized a keypoint-driven motion representation to models scene motion by leveraging the locality and consistency of motion. This approach significantly reduces storage requirements through motion-sensitive keypoint selection and an adaptive motion driven mechanism. To address error accumulation over time, we further introduce an error-aware correction strategy that mitigates these error in an efficient manner. Experiments demonstrate the surpassing storage efficiency, competitive visual fidelity and rendering speed of our method.

**Limitations:** Notably, our method still remains a few limitations. First, as the first frame serves as the foundation for subsequent frame updates, poor initialization would lead to error propagation and degraded performance. Developing a robust and efficient initialized strategy for first frame could further improve the visual quality and storage efficiency of online FVV. Second, our method relies on the dense view videos as inputs, which is expensive for practical applications. Future work will explore extending the framework to sparse-view or monocular inputs for real-world scenarios. Additionally, this method does not fully consider the training time in the encoding stage, leaving room for further improvements in training efficiency. In future work, we aim to design a practical solution on novel applications, such as 3D video conference and volumetric live streaming, providing viewers with immersive and interactive experiences.

## Acknowledgements

This work was supported by the National Natural Science Foundation of China (Grant No.62031013), the Guangdong Province Key Construction Discipline Scientific Research Capacity Improvement Project (Grant No.2022ZDJS117), Engineering Technology R&D Center of Guangdong Provincial Universities (Grant No.2024GCZX004) and the Pengcheng Laboratory.

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

## Appendix

We provide more material to supplement our main paper. This appendix first introduces more implementation details in Sec. A. Then, we provide additional experimental results in Sec. B, and broader impact in Sec. C.

## A  More Implementation details

**Training:** Our code is based on the open-source code of 3DGStream [2]. On both N3DV and MeetRoom dataset, we utilize COLMAP [50] to generate the initial point cloud and vanilla 3DGS [17] to initialize the Gaussians for 3000 epochs at first frame. Subsequently, our ComGS reconstructs the non key frames for 150 epochs and key frames for 1000 epochs. For the balance of visual quality and storage requirements, we set spherical harmonics (SH) degree to 1. During training, the learning rate for Gaussian attributes is set to 0.002, for the attributes of the adaptive influence region to 0.02, and for the learnable mask $m_i$ to 0.01. Other learning rates follow the setting of 3DGStream [2].

**Compression:** For the reconstruction process, the uncompressed Gaussian attributes and their residuals have substantial memory requirements. We employ quantization and entropy coding to further compress them. Specifically, for the first frame reconstruction, we apply 16-bit quantization to the position attributes due to their higher sensitivity, while the other attributes are quantized to 8 bits. For the correction in key frame reconstruction, we quantize all attribute residuals using 8 bits. Notably, the attributes of a keypoint play a crucial role in guiding the motion of nearby non-keypoints. As a result, even minor quantization errors in keypoints may be amplified throughout the scene. To preserve modeling accuracy, we thus refrain from quantizing keypoint attributes. Finally, we deliver these quantized values to entropy coding [55].

**Datasets: (1) Neural 3D Video (N3DV) dataset** [1] comprises of six indoor scenes captured by a multi-view system of 18 to 21 cameras at a resolution of 2704×2028 and 30 FPS. Following the previous works [2, 1, 8], we downsample the videos by a factor of 2 for training and testing and employ the central view for testing view. **(2) MeetRoom dataset** [37] is captured by a 13-camera multi-view system, including four dynamic scenes at 1280×720 resolution and 30 FPS. The center reference camera is also used for testing. As the aforementioned two datasets contain 300 frames, we also conduct long video reconstruction evaluation on the *Flame Salmon* scene with 1200 frames from the N3DV dataset. We perform distortion for this dataset following the settings of the 3DGS [17] to improve the reconstruction quality.

Table 6: Quantitative results of the random access version on N3DV dataset [1].

| Metric | Coffee Martini | Cook Spinach | Cut Beef | Flame Salmon | Flame Steak | Sear Steak |
|---|---|---|---|---|---|---|
| PSNR (dB ↑) | 28.52 | 32.31 | 32.97 | 29.19 | 33.01 | 33.51 |
| Storage (KB ↓) | 177.4 | 115.3 | 119.3 | 168.6 | 114.7 | 105.3 |

Table 7: Ablation study on Number of keypoints.

| #Keypoints | 50 | 100 | 200 | 300 | 400 | 500 |
|---|---|---|---|---|---|---|
| PSNR (dB ↑) | 31.77 | 31.85 | 31.87 | 31.84 | 31.86 | 31.80 |
| Storage (KB ↓) | 44.4 | 46.2 | 50.1 | 50.2 | 54.4 | 57.3 |

## B  Additional Experimental Results

### B.1  Random Access

Random access is crucial for video streaming and interactive user experiments. However, existing online FVV reconstruction methods [2, 13, 14] rely on the Gaussian points of the previous frame during each current frame reconstruction, thus only supporting forward playback from the first frame.

In contrast, our method enables random access by simply modifying a small part of the system configuration. Specifically, compared to the original setting, we instead reconstruct non-key frames

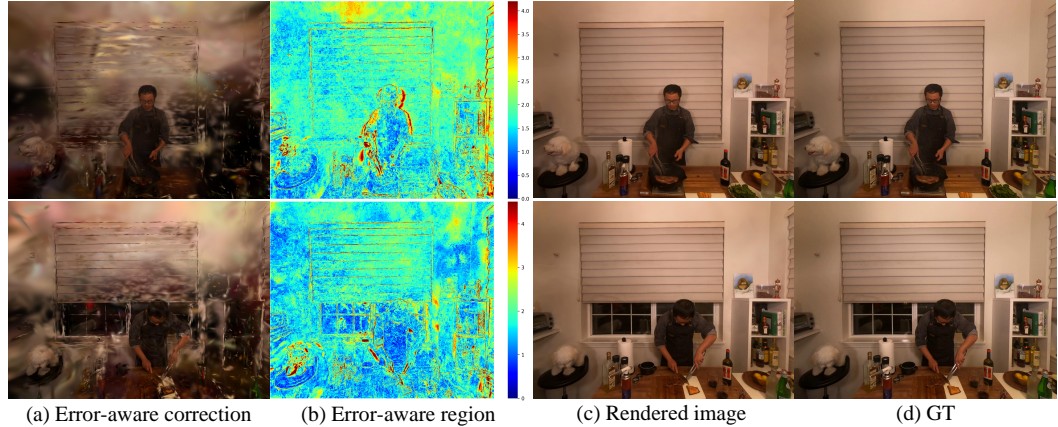

|(a) Error-aware correction | (b) Error-aware region | (c) Rendered image | (d) GT |

Figure 7: (a) Visualization of error-aware Gaussians. (b) Visualization of error regions between key frame and previous frame. (c)(d) Comparison on rendered images and original images.

using the keypoints from their nearest preceding key frame. Key frames are reconstructed based on the previous key frame using error-aware correction. Additionally, to further decouple key frames from earlier ones, we introduce periodic I-frames (e.g., every 60 frames), in which all Gaussian primitives are either saved or re-optimized independently. With this adaptation, accessing a specific frame only requires access to its nearest preceding key frame and the associated keypoints, making random access feasible. Tab. 6 presents the quantitative results of the random access version.

Table 8: Ablation study on Group of Frames.

| #Frames | 2 | 5 | 10 | 15 | 20 |
|---|---|---|---|---|---|
| PSNR (dB ↑) | 32.12 | 32.01 | 31.87 | 31.78 | 31.66 |
| Storage (KB ↓) | 108.3 | 66.6 | 50.1 | 43.2 | 40.0 |

## B.2 More Ablation Study

In this section, we further investigate the hyperparameters and analyze the impact of the proposed components on N3DV [1] dataset, to achieve a balance between performance and efficiency.

**Effect of the keypoint numbers**: To investigate the impact of the number of keypoints on reconstruction quality and compression efficiency, we conduct an ablation study by varying the number of keypoints from 50 to 500. As shown in Tab. 7, the reconstruction performance peaks when using 200 keypoints. This observation aligns with the nature of dynamic scenes, where motion typically occurs in a limited spatial region. Using 200 keypoints is sufficient to capture these areas for effective reconstruction. Increasing the number of keypoints beyond this leads to redundant or incorrect representation in static regions. Therefore, using 200 keypoints strikes a good balance between performance and storage, and is adopted as the default configuration in our method.

**Effect of group of frames**: We evaluate how the size of the Group of Frames (GoF) affects reconstruction quality and storage, as shown in Tab. 8. These results indicate that shorter GoFs can better handle non-rigid motions and novel objects, which are difficult to be reconstructed by keypoint-driven motion. Larger GoFs exploit temporal redundancy for better compression, but may accumulate errors in the presence of motion and scene changes. In our setting, we use GoF = 2 as our *large* model for high-fidelity reconstruction, and GoF = 10 as our *small* model for compact representation.

**Effect of error-aware correction**: We explore the effect of the parameter $\lambda_{\text{error}}$ on reconstruction quality and storage, as shown in Tab. 9. While a larger $\lambda_{\text{error}}$ improves compression by focusing only on perceptually salient errors, it may overlook subtle regions, which leads to degraded reconstruction. In contrast, smaller values retain more points, which helps suppress error accumulation across frames, albeit with higher storage costs.

Table 9: Effect of $\lambda_{error}$ on reconstructed quality and storage.

| $\lambda_{error}$ | 0 | 0.0001 | 0.001 | 0.01 |
|---|---|---|---|---|
| PSNR (dB ↑) | 31.91 | 31.91 | 31.87 | 31.79 |
| Storage (KB ↓) | 183.0 | 96.3 | 50.1 | 29.2 |

Table 10: **Per-scene quantitative results on the N3DV dataset**. Offline and online methods are separated for clarity.

| Method | Coffee Martini | | Cook Spinach | | Cut Beef | |
|---|---|---|---|---|---|---|
| | PSNR (dB ↑) | Storage (MB ↓) | PSNR (dB ↑) | Storage (MB ↓) | PSNR (dB ↑) | Storage (MB ↓) |
| KPlanes [56] | 29.99 | 1.0 | 32.60 | 1.0 | 31.82 | 1.0 |
| NeRFPlayer [41] | 31.53 | 18.4 | 30.56 | 18.4 | 29.35 | 18.4 |
| HyperReel [52] | 28.37 | 1.2 | 32.30 | 1.2 | 32.92 | 1.2 |
| 4DGS [4] | 28.33 | 29.0 | 32.93 | 29.0 | 33.85 | 29.0 |
| 4D-GS [8] | 27.34 | 0.3 | 32.46 | 0.3 | 32.49 | 0.3 |
| Spacetime-GS [6] | 28.61 | 0.7 | 33.18 | 0.7 | 33.52 | 0.7 |
| E-D3DGS [11] | 29.33 | 0.5 | 33.19 | 0.5 | 33.25 | 0.5 |
| StreamRF [37] | 27.84 | 31.84 | 31.59 | 31.84 | 31.81 | 31.84 |
| 3DGStream [2] | 27.75 | 7.80 | 33.31 | 7.80 | 33.21 | 7.80 |
| QUEEN-l [13] | 28.38 | 1.17 | 33.40 | 0.59 | 34.01 | 0.57 |
| ComGS-s (ours) | 28.63 | 0.058 | 32.94 | 0.047 | 33.30 | 0.051 |
| ComGS-l (ours) | 28.76 | 0.154 | 33.26 | 0.094 | 33.53 | 0.104 |
| | **Flame Salmon** | | **Flame Steak** | | **Sear Steak** | |
| | PSNR (dB ↑) | Storage (MB ↓) | PSNR (dB ↑) | Storage (MB ↓) | PSNR (dB ↑) | Storage (MB ↓) |
| KPlanes [56] | 30.44 | 1.0 | 32.38 | 1.0 | 32.52 | 1.0 |
| NeRFPlayer [41] | 31.65 | 18.4 | 31.93 | 18.4 | 29.12 | 18.4 |
| HyperReel [52] | 28.26 | 1.2 | 32.20 | 1.2 | 32.57 | 1.2 |
| 4DGS [4] | 29.38 | 29.0 | 34.03 | 29.0 | 33.51 | 29.0 |
| 4D-GS [8] | 29.20 | 0.3 | 32.51 | 0.3 | 32.49 | 0.3 |
| Spacetime-GS [6] | 29.48 | 0.7 | 33.40 | 0.7 | 33.46 | 0.7 |
| E-D3DGS [11] | 29.72 | 0.5 | 33.55 | 0.5 | 33.55 | 0.5 |
| StreamRF [37] | 28.26 | 31.84 | 32.24 | 31.84 | 32.36 | 31.84 |
| 3DGStream [2] | 28.42 | 7.80 | 34.30 | 7.80 | 33.01 | 7.80 |
| QUEEN-l [13] | 29.25 | 1.00 | 34.17 | 0.59 | 33.93 | 0.56 |
| ComGS-s (ours) | 29.31 | 0.052 | 33.42 | 0.045 | 33.59 | 0.040 |
| ComGS-l (ours) | 29.58 | 0.129 | 33.84 | 0.083 | 33.74 | 0.0704 |

Fig. 7 (a) visualizes the error-aware Gaussian points identified by error-aware correction, while (b) shows a heatmap of differences between the key frame and the previous frame, which highlights the error regions. We observe that the error-aware points in (a) align well with the high-error regions in (b), which indicates that our method effectively captures areas likely to suffer from error accumulation. Fig. 7 (c) and (d) compare our rendered images with the ground truth. The results show that our method significantly reduces artifacts in dynamic regions, confirming the effectiveness of our error-aware correction.

## B.3 More Results

To offer a more comprehensive comparison, the per-scene quantitative results are presented on N3DV [1] and MeetRoom [37] in Tab. 10 and Tab. 11, respectively. Moreover, we also provide the experimental results of existing offline and online methods in Tab. 10 as a reference. Further qualitative results with StreamRF [37] and 3DGStream [2] are indicated in Fig. 8 and Fig. 9.

# C Broader Impact

Our work is a positive technology. This method reconstructs free-viewpoint videos from multi-view 2D videos in a streaming manner, which can improve the immersive and interactive experience of viewers. As discussed in the introduction, this technology has potential to benefit various aspects of daily life, including applications in remote diagnosis and 3D video conferencing.

Table 11: **Per-scene quantitative results on the MeetRoom dataset**.

| Metrics | Discussion | Stepin | Trimming | VrHeadset |
|---------|-----------|--------|----------|-----------|
| PSNR (dB ↑) | 31.72 | 30.17 | 32.12 | 31.95 |
| Storage (KB ↓) | 37.5 | 24.2 | 27.0 | 24.5 |

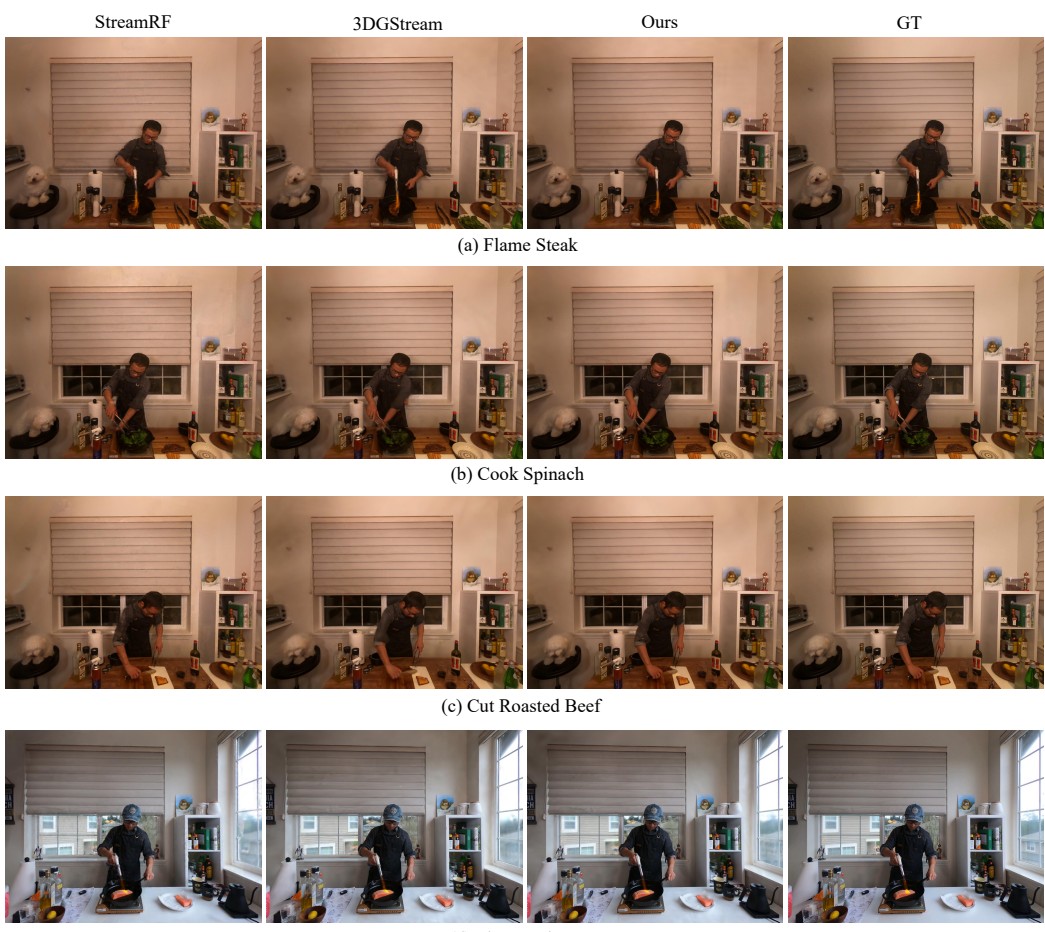

(a) Flame Steak

(b) Cook Spinach

(c) Cut Roasted Beef

(d) Flame Salmon

Figure 8: Comparison on N3DV [1] dataset.

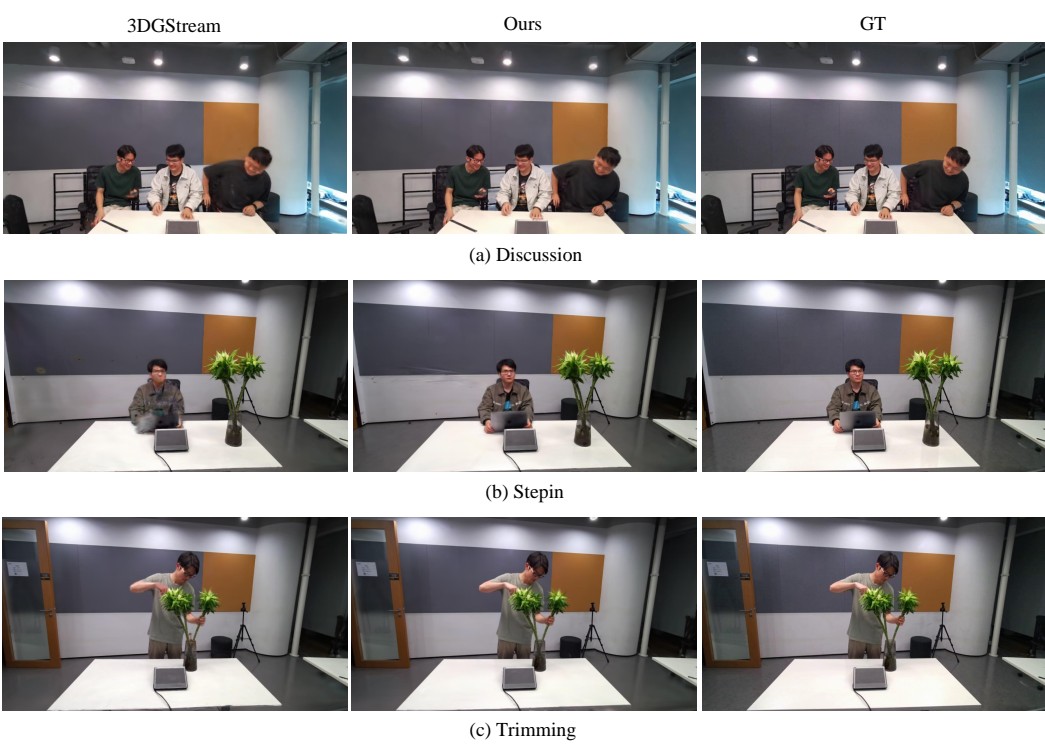

3DGStream            Ours            GT

(a) Discussion

(b) Stepin

(c) Trimming

Figure 9: Comparison on MeetRoom [37] dataset.

