# OpenReview forum: "Motion Matters: Compact Gaussian Streaming for Free-Viewpoint Video Reconstruction"
_NeurIPS.cc/2025/Conference — NeurIPS 2025 poster_

### Official Review · Reviewer_HKLW · 2025-06-30

**Clarity:** 3
**Significance:** 4
**Originality:** 3
**Rating:** 4
**Confidence:** 3

**Summary:**

Paper addresses the task of Streaming for Free-Viewpoint Video Reconstruction:

For this the frames are first grouped (GpF) with fixed size(?).
The first frame is reconstructed with 3DGS and the remaining frames of the group are modeled with deforming the Gaussians of the first frame.
Keypoints with Gaussian influence fields are used to model motion in compact efficient manner.
Error aware masking is used to only make necessary update of parameters between key frames.

**Questions:**

I am not familiar with the prior works like QUEEN and 3DGSteam. I think parts of the paper are hard to understand since the proposed methods heavily builds on these works.

Nevertheless, most of the paper is well written and I was able to understand almost all of it.
My only concern is the section "Error-Aware Corrector".
To my understanding each keyframe is reconstructed individually (or just the very first one?). How are the Gaussian's matched between keyframes?
If the reconstruction is only done once for the first keyframe, then how does the densification work? Or this there none?
Improving this part of the paper would definitively help to make the whole paper easier to understand.

If the authors address my concerns regarding that section I will consider raising the score.

**Ethical Concerns:**

["NO or VERY MINOR ethics concerns only"]

**Final Justification:**

The authors addressed my concerns in their rebuttal. Nevertheless, I believe that the section on the  error-aware corrector should be revised beyond a minor change. Therefore, I am leaving my rating at “borderline accept"

**Limitations:**

Limitations are addressed but only in appendix.

**Quality:**

3

**Strengths And Weaknesses:**

Strengths:

- The proposed method achieves SOTA compression with very good visual quality
- An extensive ablation study is performed to show the effect of each component
- good visualization of components like key point selection which helps to understand the method and gives an intuition on how it works
- easy to follow, compact mathematical notation

Weaknesses:

- Limitations are in appendix. IMO need to be moved (or short version of it) to main paper
- No 3DGS compression methods in related work
- Preliminary is missing the EWA splatting part
- Evaluation datasets are limited. What about orbital camera datasets?

---

> ### Author Rebuttal · Authors · 2025-07-31
>
> We sincerely appreciate the reviewer’s positive feedback and the careful review of our work. Below, we address each of the concerns in detail.
>
> ## Q1: Confusion on error-aware corrector.
> We sincerely apologize for the confusion caused in this section.
>
> To clarify, the keyframes in our method are not reconstructed independently. Performing independent reconstruction for every keyframe would result in significant storage overhead. In fact, only the first frame of the entire video is reconstructed and stored with its full set of Gaussian primitives—this frame serves as the **I-frame**. **The subsequent keyframes do not involve independent reconstruction**. Instead, their primary purpose is to correct the accumulated errors introduced by keypoint motion modeling. These keyframes provide periodic corrections to ensure fidelity in the online reconstruction process. To eliminate any ambiguity, we will explicitly clarify this point in the revised version.
>
> For **subsequent frame reconstruction** (i.e., non-I-frame reconstruction), our method reuses the Gaussian primitives from the previous frame as a reference. Specifically:
>
> - **For non-keyframes**, the modeling relies on motion estimation between the current and previous frame to select keypoints, and the scene is modeled based on the Gaussian primitives from the previous frame.
> - **For keyframes**, we also use the Gaussian primitives from the previous frame, but apply error-aware correction to update those primitives that contain modeling errors.
>
> Therefore, only the first frame is reconstructed independently and includes a Gaussian densification process. Subsequent frames do not involve independent reconstruction or densification.
>
> We hope this explanation resolves the confusion. If there are any remaining concerns, we would be happy to discuss further. We will also revise the corresponding section in revised version to make this process more clearly stated.
>
> ## Q2: Limitations in appendix.
> Thank you for your valuable suggestion. We agree with the importance of presenting the limitations in the main paper. In the revised version, we will include a condensed version of the limitations discussion in the conclusion section. We also present the summarized version below in this rebuttal:
>
> **Limitations.** While our method achieves notable improvements, several limitations remain. First, the quality of the initial frame significantly impacts subsequent performance, and poor initialization may lead to error accumulation. Second, although our approach focuses on compact representation, the training efficiency of the encoding stage can be further optimized. Lastly, our method currently requires dense-view video input, which may limit its practicality; future work will explore extensions to sparse-view or monocular settings.
>
> ## Q3: 3DGS compression methods in related work.
> We appreciate the reviewer’s valuable suggestion. Discussing 3DGS compression methods in the related work section will indeed enhance the completeness of our paper.
>
> Existing 3DGS compression approaches can be broadly categorized into two types:
> - Reducing the number or precision of Gaussian parameters。
> - Exploiting spatial structural correlations to reduce the overall size of 3DGS.
>
> The first category includes techniques such as removing unimportant Gaussians, pruning spherical harmonic (SH) coefficients, and applying vector quantization to compress the parameter representation. The second category focuses on structured predictions of unstructured 3D points, typically through multi-resolution grids or triplane-based representations, which effectively reduce the storage requirements of 3DGS.
>
> We will incorporate a discussion of these 3DGS compression techniques in the revised version.
>
> ## Q4: EWA splatting.
> We appreciate the reviewer for the insightful suggestion. We agree that EWA splatting[E] is indeed relevant, as it provides the theoretical foundation for projecting 3D Gaussians into 2D image space using Jacobian of the affine approximation of the projective transformation. We will incorporate a discussion of EWA splatting and cite the relevant literature in the revised version to clarify the connection.
>
> [E] Zwicker M, Pfister H, Van Baar J, et al. Ewa volume splatting[C]//Proceedings Visualization, 2001. VIS'01. IEEE, 2001: 29-538.
>
> ## Q5: Limited in evaluation dataset.
> We appreciate the reviewer’s concern regarding dataset selection.
>
> In our experiments, we use N3DV and MeetRoom, both of which are widely adopted in the online Free-Viewpoint Video (FVV) reconstruction community and consist of multi-view video sequences. In contrast, datasets captured with orbital camera setups are less commonly used in online FVV research, as they typically do not provide the temporally synchronized multi-view video inputs required for frame-by-frame training.
>
> To further demonstrate the generalizability and effectiveness of our method, we have additionally conducted experiments on the Google Immersive dataset, which is another widely used benchmark for free-viewpoint rendering. We provide the quantitative results below, and will include them in the revised version of the paper.
>
> **Tab. R11:** Quantitative results on the Google immersive dataset.
>
> | Method     | PSNR| Size(MB) |
> |------------|--------|--------|
> | 3DGStream      | 25.18  | 8.83  |
> | Ours       | 27.24  |  0.16 |
>
> We hope this response clarifies your concerns. Thank you again for your valuable comments and encouraging review.

---

> ### Author Response · Authors · 2025-08-05
>
> Dear Reviewer HKLW,
>
> Thank you for your careful review and positive feedback on our work. We appreciate the time you’ve taken to evaluate our submission. In our rebuttal, we addressed your concerns regarding the "Error-Aware Corrector" section and provided additional discussion on 3DGS compression and EWA splatting. We also included results on an additional dataset. If you’ve had a chance to review our rebuttal, we’d be grateful to know whether our explanations have addressed your concerns. If so, we would sincerely appreciate it if you could consider updating your evaluation accordingly.
>
> Thank you for your hard work and support!
>
> Best regards,
>
> The authors of Paper 20548

---

> ### Author Response · Authors · 2025-08-07
> **Thank you for your careful review!**
>
> Dear Reviewer HKLW,
>
> Thank you once again for your valuable comments and for taking the time to read our rebuttal.
>
> We would greatly appreciate it if you could let us know whether our response **has fully addressed your concerns**, and whether it might lead you to reconsider your evaluation. If there are still unresolved concerns, we would be more than happy to continue the discussion with you.
>
> Thank you again for your thoughtful and constructive review.
>
> Best regards,
>
> The authors of Paper 20548

---

> ### Author Response · Authors · 2025-08-08
>
> Dear Reviewer HKLW,
>
> Thank you again for your thoughtful feedback to our work.
>
> In our rebuttal, each of the concerns you raised has been carefully addressed. In particular, a detailed explanation has been provided regarding the error-aware corrector, which appeared to be your main concern. This clarification will also be reflected in the revised manuscript. If there are still aspects that remain unclear, we would sincerely appreciate the opportunity to clarify them further.
>
> The compression performance of our method was found to be impressive, and the reviewers agreed that it provides new insights that may benefit future research and applications. Every effort has been made to address the concerns raised by the reviewers through detailed clarifications and revisions. These revisions have received positive responses from all reviewers, with one even raising their score after reviewing our responses.
>
> We would be sincerely grateful if you could kindly reconsider your evaluation in light of these contributions.
>
> Thank you for your hard work and support!
>
> Best regards,
>
> The authors of Paper 20548

---

### Official Review · Reviewer_w6Sx · 2025-07-02

**Clarity:** 3
**Significance:** 3
**Originality:** 3
**Rating:** 4
**Confidence:** 4

**Summary:**

The paper proposes the ComGS framework, which optimizes online free-viewpoint video (FVV) reconstruction and transmission through keypoint-driven motion modeling. The core innovations include:
1. Motion-sensitive keypoint selection, which locating dynamic regions based on viewpoint spatial gradient differences;
2. Adaptive motion-driven mechanism through propagating keypoint motion to neighboring Gaussian points via a spatial influence field;
3. Error-aware correction strategy which selectively optimizes error-prone regions in keyframes.
Experiments show that ComGS achieves significant performance improvements and real-time rendering speed.

**Questions:**

1. While Section 3.4 examines selective residual updates, it lacks in-depth discussion regarding the handling of newly emerging objects. Can you further explain how this article addresses this issue?

2. The current methods for FVV are largely confined to existing benchmark datasets and lack validation on challenging scenarios involving non-rigid objects, varying illumination conditions, and rapid motions. Can the method proposed in this article be applied to such scenarios?

**Ethical Concerns:**

["NO or VERY MINOR ethics concerns only"]

**Final Justification:**

While minor points remain, the authors' revisions have sufficiently improved the manuscript to warrant acceptance.

**Limitations:**

Yes.

**Quality:**

3

**Strengths And Weaknesses:**

**Strengths**

This paper proposes many effective methods to improve performance of FVV task, which includes:
1. The keypoint selection strategy effectively focuses on dynamic regions;
2. The spatial influence field replaces traditional KNN, adaptively accommodating varying motion scales;
3. Error-aware correction effectively identifies the Gaussians need to be corrected.

Comprehensive comparisons with online/offline methods on mainstream datasets and ablation studies validate the performance of the proposed method and the necessity of each module.

**Weakness**

1. Relies on the rigid motion assumption, which may fail for non-rigid deformations. Besides, fixed number of keypoints is not adapted to varying motion complexity across scenes.
2. For quality results, only PSNR is reported.  Including more evaluation (SSIM, LPIPS, etc.)metrics can better reflect the performance of the proposed method in the paper.

---

> ### Author Rebuttal · Authors · 2025-07-31
>
> We sincerely appreciate the reviewer for the careful and thoughtful review of our work. Below, we address the concerns and limitations you pointed out, and provide our corresponding responses and solutions.
>
> ## Q1: Failure for non-rigid motion
> We sincerely appreciate your valuable feedback. We have indeed considered this issue during the development of our method. To effectively model complex motion, we propose the Adaptive Motion-Driven Mechanism and the Error-Aware Corrector, which work together to address this challenge.
>
> Specifically, in non-keyframes, our method models motion by leveraging the spatial influence fields of multiple keypoints. We allocate a sufficient number of keypoints to ensure that even complex or non-rigid motions can be accurately captured through their combined influence. However, this approximation can introduce errors. To address this, during keyframe stages, we apply the Error-Aware Corrector to identify and correct Gaussian primitives with accumulated errors caused by non-rigid or more complex motion patterns.
>
> Since our method aims to minimize the storage required for FVV, there is an inherent trade-off in reconstruction accuracy. Nevertheless, this can be mitigated by reducing the interval between keyframes, allowing the method to adaptively retain higher fidelity when needed.
>
> ## Q2: Fixed numbers of keypoints.
> In our experimental setup, we used a fixed number of keypoints (200) across all datasets and videos. Our experiments and visualizations show that this number is sufficient to effectively cover motion regions and model dynamic changes in various scenes.
>
> We also greatly appreciate the reviewer’s insightful suggestion. We believe that exploring a strategy to adaptively determine the number of keypoints, based on the scale of motion in the scene, could lead to more efficient keypoint allocation and better adaptability to different motion complexities. This is a promising direction we plan to investigate in future work.
>
> ## Q3: Lack of more evaluation metric.
> We greatly appreciate the reviewer’s insightful suggestion, as we fully agree that incorporating more evaluation metrics is essential for providing a more comprehensive and balanced assessment of the performance and effectiveness. We include additional evaluation results below and will further incorporate these metrics into the revised version to ensure a more thorough and rigorous evaluation of our approach.
>
> **Tab. R7:** Quantitative results on the N3DV dataset.
>
> | Method     | PSNR ↑ | SSIM ↑ | LPIPS ↓ |
> |------------|--------|--------|---------|
> | 3DGStream      | 31.67  | 0.9410  | 0.1400   |
> | Ours       | 31.87  |  0.9434 |  0.1320  |
>
> **Tab. R8:** Quantitative results on the MeetRoom dataset.
>
> | Method     | PSNR ↑ | SSIM ↑ | LPIPS ↓ |
> |------------|--------|--------|---------|
> | 3DGStream      | 30.79  | 0.9498  | 0.1882   |
> | Ours       | 31.49  |  0.9548 |  0.1709  |
>
> ## Q4: Discussion on newly emerging objects.
> Similar to the discussion in Q1, the modeling of newly emerging objects is handled through the error-aware corrector applied at key frames. Although our method does not introduce new Gaussian primitives specifically for the newly appeared objects, the large number of redundant Gaussians[D] retained in the scene provides sufficient representation capacity to model such objects effectively.
>
> As discussed in Q1, minimizing the storage may lead to a drop in reconstruction quality. However, this issue can be alleviated by reducing the keyframe interval, which allows for more frequent correction and better adaptation to scene changes. Despite the lack of additional Gaussian primitives for new objects, our method still demonstrates robust performance in handling them.
>
> For example, our experiments on the stepin (MeetRoom dataset) where a new person enters the meeting room. Our method achieves superior reconstruction quality, both in quantitative metrics and visual results. The quantitative results are as follows:
>
> **Tab. R9:** Quantitative results on the stepin.
>
> | Method     | PSNR ↑ | SSIM ↑ | LPIPS ↓ |
> |------------|--------|--------|---------|
> | 3DGStream      | 28.35  | 0.9351  | 0.2084   |
> | Ours       | 30.04  |  0.9451 |  0.1862  |
>
> [D] Wang H, Zhu H, He T, et al. End-to-end rate-distortion optimized 3d gaussian representation[C]//European Conference on Computer Vision. Cham: Springer Nature Switzerland, 2024: 76-92.
>
> ## Q5: Lack of validation on challenging scenarios.
> Due to the capability of our method to model complex motion and to correct modeling errors, it is well-suited to handle challenging scenarios such as non-rigid objects, varying illumination conditions, and rapid motions.
>
> To further validate the effectiveness of our approach in such settings, we conducted additional experiments on the Google Immersive dataset, which features highly dynamic scenes with extensive motion and newly appearing elements such as flames. Our method achieves comparable reconstruction quality on this dataset. The experimental results are presented below and will be included in the revised version:
>
> **Tab. R10:** Quantitative results on the Google immersive daatset.
>
> | Method     | PSNR| Size(MB) |
> |------------|--------|--------|
> | 3DGStream      | 25.18  | 8.83  |
> | Ours       | 27.24  |  0.16 |

---

> ### Author Response · Authors · 2025-08-05
>
> Dear Reviewer w6Sx,
>
> Thank you again for your thorough review and your enthusiasm for our work. We have addressed your thoughtful questions and added more results in our rebuttal. We await your reply and are more than wiling to further discuss and clarify any addtional questions you may have. Alternatively, if you find that some of your concerns have been addressed, we kindly request that you consider updating your evaluation accordingly, we would be very very very grateful!
>
> Sincerely appreciate for your hard work and support!
>
> Best regards,
>
> The authors of Paper 20548

---

> > ### Comment · Reviewer_w6Sx · 2025-08-06
> >
> > Thanks for the authors' effort. Their explanation is reasonable, and I will keep my accept rating.

---

> > > ### Author Response · Authors · 2025-08-07
> > >
> > > Thank you very much for your positive feedback and for taking the time to consider our rebuttal carefully!

---

### Official Review · Reviewer_ffQB · 2025-07-02

**Clarity:** 3
**Significance:** 2
**Originality:** 2
**Rating:** 4
**Confidence:** 3

**Summary:**

This paper proposes a compact and motion-aware Gaussian Splatting framework for dynamic scene modeling. Existing methods suffer from excessive storage requirements due to redundant motion modeling. ComGS addresses this by (1) identifying sparse keypoints (≈200) in motion regions using viewspace gradient differences, avoiding redundant motion modeling; (2) propagating keypoint motion to neighboring Gaussian points via a spatial influence field, enabling shared motion representation and reducing storage; (3) periodically refines only error-prone Gaussian points to mitigate error accumulation without redundant updates. Experiments on N3DV and MeetRoom datasets show ComGS achieves a 159× storage reduction over 3DGStream and 14× over SOTA method QUEEN, while maintaining competitive PSNR and rendering speed.

**Questions:**

1. The motion-sensitive keypoint selection uses a gradient difference threshold. How is this threshold determined? Does it adapt to scene complexity?

2. The paper uses 200 keypoints. What is the impact of varying this number (e.g., 100 vs. 300) on storage and fidelity? Is there an optimal range?

3. The words ''spatial influence field'' is confusing. Is it just the covariance?

Currently my suggestion is boarderline, my concerns and questions are listed above. The comparison with Representing Long Volumetric Video with Temporal Gaussian Hierarchy is necessary, which also focuses on the efficiency of reconstructing dynamic multi-view videos.

**Ethical Concerns:**

["NO or VERY MINOR ethics concerns only"]

**Final Justification:**

I have read the responses and my concerns have been addressed. The discussions and comparisons metioned above should be included in the final version.

**Limitations:**

Yes

**Quality:**

2

**Strengths And Weaknesses:**

Strength
1. By focusing on motion regions and shared keypoint-driven motion, ComGS drastically reduces per-frame storage (0.049 MB/frame vs. 7.8 MB/frame for 3DGStream), enabling real-time transmission.
2. ComGS maintains high visual quality (PSNR: 33.61 dB) and fast rendering (up to 147 FPS), outperforming baselines in both metrics.
3. The ablation studies are adequate.

Weakness
1. The paper lacks the discussion and comparison with highly related works, i.e. Representing Long Volumetric Video with Temporal Gaussian Hierarchy[1], FreeTimeGS[2].

[1] Xu, Zhen, et al. "Representing long volumetric video with temporal gaussian hierarchy." ACM Transactions on Graphics (TOG) 43.6 (2024): 1-18.
[2] Wang, Yifan, et al. "FreeTimeGS: Free Gaussian Primitives at Anytime Anywhere for Dynamic Scene Reconstruction." Proceedings of the Computer Vision and Pattern Recognition Conference. 2025.

2. Keypoint-driven motion works seems similar to KNN-based method like SC-GS, the ablation study for KNN and Adaptive control shows that the performance is comparable.

3. Performance depends on key frame spacing (e.g., ComGS-s uses s=10, ComGS-l uses s=2). Optimal intervals may vary across scenes, requiring manual tuning.

4. If keypoints fail to accurately identify motion regions (e.g., in low-contrast scenes), motion propagation could be inaccurate, degrading quality.

5. The rendering speed drops a lot compared to 3DGStream, QUEEN.

---

> ### Author Rebuttal · Authors · 2025-07-31
>
> We appreciate the reviewer for the careful and thoughtful review of our work. In the following we address the concerns in the order as pointed out by the reviewer.
>
> ## Q1: Lack of comparison and discussion on highly related work.
> We sincerely appreciate the reviewer for pointing out the lack of discussion and comparison with highly relevant works, and for recommending two high-quality papers that are closely related to our topic.
>
> Although both methods are designed for offline reconstruction, we have greatly benefited from studying these works. Specifically, [B] proposes a multi-level 4D Gaussian representation that captures scene content at different levels of detail, which enables efficient reconstruction of long FVV sequences. [C], on the other hand, introduces a motion function for each Gaussian primitive, allowing it to handle complex dynamic scenes and achieve impressive reconstruction quality.
>
> We will incorporate a detailed discussion of these two works in the Related Work section of our revised version, and will also include quantitative comparisons in the Experiments section. As a preliminary step, we provide the following quantitative results for comparison:
>
> **Tab. R4:** Quantitative results on the flame salmon (1200 frames).
>
> | Method     | PSNR  | Size (MB) |
> |------------|--------|--------|
> | [B]      | 29.44  | 0.075  |
> | Ours       | 29.58  | 0.054  |
>
> **Tab. R5:** Quantitative results on the N3DV dataset.
>
> | Method     | PSNR  | Size (MB) |
> |------------|--------|--------|
> | [C]      | 32.97  | 0.137  |
> | Ours-L       | 32.12  | 0.106  |
>
> While our method does not yet match the reconstruction quality of [C], our focus is on online reconstruction with high storage efficiency, which addresses a different and complementary aspect of the problem. We believe that the insights from [B] and [C] will inspire future improvements in our method's representation capacity, and we plan to explore these directions in subsequent work.
>
> [B] Xu, Zhen, et al. "Representing long volumetric video with temporal gaussian hierarchy." ACM Transactions on Graphics (TOG) 43.6 (2024): 1-18.
> [C] Wang, Yifan, et al. "FreeTimeGS: Free Gaussian Primitives at Anytime Anywhere for Dynamic Scene Reconstruction." Proceedings of the Computer Vision and Pattern Recognition Conference. 2025.
>
> ## Q2: Confusion on adaptive motion-driven mechanism.
> **Similar to KNN-based works:** We sincerely apologize for the confusion regarding the distinction between our method and KNN-based approaches. We provide a detailed explanation below.
>
> KNN-based methods typically compute neighbor weights solely based on Euclidean distances, without adapting to the motion scale in the scene. In contrast, our method is designed for online reconstruction and leverages a spatial influence field that is learned in a motion-adaptive manner. This enables our model to dynamically adjust the spatial extent of each keypoint’s influence according to the underlying motion scale, leading to more accurate motion modeling.
>
> As the reviewer correctly pointed out, the results in Table 5 show that our method performs comparably to the KNN-based baseline. This is because both methods in this ablation study include our third proposed component—error-aware correction—which effectively mitigates the error accumulation typically observed in KNN-based approaches. However, even under this setting, our method still outperforms the KNN-based approach by 0.48 dB in PSNR, demonstrating the effectiveness of our design.
>
> **Confusion on spatial influence field:** Our spatial influence field is parameterized by a learnable covariance matrix. By learning this matrix, our method is able to control both the selection of sptially neighboring points and the influence strength of each keypoint on its neighbors. This formulation allows the model to adaptively adjust the influence region to suit different motion scales in the scene. Therefore, we refer to this learned control region as a spatial influence field.
>
> ## Q3: Key frame spacing.
> We thank the reviewer for the insightful comment. We have indeed considered this issue. Currently, the interval between keyframes in our method is adjusted solely based on the target compression rate, and is therefore fixed across all datasets.
>
> As the reviewer rightly pointed out, the optimal keyframe interval is likely scene-dependent. We agree that a fixed interval based only on compression rate is a simple heuristic, and a more adaptive strategy could achieve a better balance between reconstruction quality and storage efficiency. In future work, we plan to explore dynamically adjusting the interval based on factors such as scene dynamics or reconstruction error.
>
> ## Q4: Confusion on keypoint selection.
> **Indentifying motion region:** Currently, publicly available datasets lack of the low-contrast scenes. In our work, we conduct extensive evaluations on several widely used benchmark datasets, demonstrating strong motion region identification performance. Notably, the N3DV dataset contains relatively subtle and fine-grained motions and our method is still able to effectively detect and localize these motion regions. In addition, we also evaluate our method on the Google Immersive dataset, which contains scenes with larger and more complex motion. Our method successfully extracts meaningful motion keypoints in these challenging scenarios as well, further demonstrating its effectiveness across a wide range of motion magnitudes.
>
> Our keypoint selection is based on the gradient of Gaussian primitives in dynamic regions across frames. This mechanism is driven by inter-frame changes rather than absolute contrast, and therefore, we believe it would remain effective even in low-contrast scenes.
>
> **Keypoint selection based on gradient difference:** We apologize for the confusion caused. In our method, a dynamic significance score is computed for each Gaussian point based on **the gradient difference** across frames (as defined in Eq. 5). This score reflects the temporal variation of each point, allowing us to identify regions with significant motion. We then rank all points based on these scores and select **the top-k** points (where k is the number of keypoints) as the keypoints for the current frame.
>
> Our experiments and ablation studies demonstrate that selecting 200 keypoints is sufficient to cover most motion regions in the scene and to capture the underlying dynamics effectively.
>
> **Numbers of keypoints:** We thank the reviewer for raising this important question. We have also considered how to set the number of keypoints, and we conducted an ablation study on this in the supplementary material. For clarity, we include the results in our rebuttal:
>
> **Tab. R6:** Ablation study on numbers of keypoints.
>
> | #Keypoints     | 50 | 100 | 200 | 300 | 400 | 500 |
> |------------|--------|--------|---------|--------------|--------------|--------------|
> | PSNR      | 31.77  | 31.85  | 31.87 | 31.84 | 31.86 | 31.80 |
> | Size (KB)  | 44.4  | 46.2  | 50.1 | 50.2 | 54.4 | 57.3 |
>
> The results show that selecting around 200 keypoints provides the best trade-off between reconstruction quality and storage cost. Interestingly, we observed that increasing the number of keypoints beyond this range does not improve performance, and in some cases, even slightly degrades it. We hypothesize that assigning too many keypoints may overfit simple motions, making them appear more complex than they actually are, which ultimately reduces the accuracy of motion modeling.
>
> ## Q5: Rendering speed.
> In this work, our primary goal is to minimize storage consumption for online FVV, thereby reducing storage and transmission requirements—an important and challenging issue in this field. Although reducing storage introduces some compromises in rendering speed, we do not consider this a fundamental limitation. Below, we explain the reasons behind the slowdown and provide corresponding solutions to address it.
>
> In particular, our implementation of Eq.6 and Eq.8 involves computations over a large number of background Gaussian primitives in influence weight matrix (size $\approx$ [200000, 200], which introduces significant redundancy and slows down rendering. However, we believe this can be addressed with the following strategies: (1) **Pruning:** Introduce a classifier to coarsely separate foreground and background Gaussian primitives, and compute $w$ only for the foreground Gaussian primitives, significantly reducing its size and may yield a 2-3× speed up. (2) **Sparse:** Note that $w$ is extremely sparse (about 99.85% of values are zero); leveraging sparse matrix techniques could greatly improve both training and rendering speed by 3-4×. (3) **Parallel acceleration:** Our adaptive motion-driven mechanism is inspired by 3DGS, which benefits from distributed optimization and GPU-accelerated algorithms—these techniques are also applicable to keypoints training.
>
> By integrating the proposed solutions along with engineering optimization, we estimate that the computational cost associated with Eq.6 and Eq.8 can be accelerated by at least 10×. This optimization would lead to a rendering speed comparable to QUEEN.
>
> We plan to incorporate these improvements in future work to enhance the rendering efficiency.

---

> ### Author Response · Authors · 2025-08-05
>
> Dear Reviewer ffQB,
>
> Thank you again for your thorough review and your positive comments on our work. In our rebuttal, we clarified the concerns you raised about our work and added a discussion and comparison with high related studies. We would be grateful if you could read our response and let us know if you have additional questions. Your insights and replies are highly appreciated. We would also be grateful that you could reconsider the evaluation rating if you feel the concerns have been sufficiently addressed.
>
> Sincerely appreciate for your hard work and support!
>
> Best regards,
>
> The authors of Paper 20548

---

> > ### Comment · Reviewer_ffQB · 2025-08-07
> > **Official Comment of Reviewer ffQB**
> >
> > Thanks for authors' detailed responses. I have read the responses and my concerns have been addressed. The discussions and comparisons metioned above should be included in the final version.

---

> ### Author Response · Authors · 2025-08-07
>
> Dear Reviewer ffQB,
>
> Thank you again for your thoughtful feedback and follow-up response to our rebuttal.
>
> All reviewers recognized our proposed keypoint-driven motion representation as novel and interesting. They also acknowledged the effectiveness of the adaptive motion-driven mechanism, which adjusts well to different motion scales and enables accurate motion control. The compression performance of our method was found to be impressive, and the reviewers agreed that it offers new insights that may be valuable for future research and applications.
>
> Every effort has been made to address the concerns raised by the reviewers. In response to commonly shared issues—such as the rendering speed trade-off—a more in-depth analysis was conducted, and several reasonable and practical solutions were introduced. The discussion and comparison with closely related work were also strengthened, and the clarity of the presentation was improved accordingly. These revisions has received positive responses from all the reviewers, with one even raising the score after reviewing the clarifications.
>
> We would be sincerely grateful if you could kindly reconsider your evaluation in light of these contributions.
>
> Thank you again for your time and valuable comment.
>
> Best regards,
>
> The authors of Paper 20548

---

### Official Review · Reviewer_JVLS · 2025-07-03

**Clarity:** 2
**Significance:** 3
**Originality:** 3
**Rating:** 4
**Confidence:** 4

**Summary:**

This paper introduces **Compact Gaussian Streaming (ComGS)**, a keypoint-driven method for novel view synthesis of dynamic scenes, applied to free-viewpoint video reconstruction.
In the first frame, a static model is trained, and the proposed **Motion-Sensitive Keypoint Selection** identifies which Gaussians should serve as keypoints using the next frame's image. For non-keyframes, scene motion is driven by these keypoints, significantly reducing data storage since only the motion of **N keypoints** (much fewer than all Gaussians) with 14 parameters each is used.
To address **accumulated errors**, a new keyframe is introduced every **s frames**, using **error-aware correction**, which updates Gaussians with significant errors. The residual of updated Gaussians will be transmitted.
Experiments show that ComGS achieves **0.05-0.1 MB/frame storage**, **competitive rendering quality**, and a **7-14x compression ratio** compared to existing SOTA work, QUEEN.

**Questions:**

1. **Training time and rendering speed**: It would be helpful to understand why the training time and rendering speed are impacted. While we understand that the increased complexity may contribute to the performance tradeoff, could you clarify why this is acceptable in the context of the method's goals?
2. **Support for random access**: Does the proposed method support random access, similar to conventional video playback, where users can start viewing from any arbitrary time point? Based on the description, it seems that decoding subsequent keyframes depends on previous non-keyframes, which might limit random access. If this is the case, could you share your thoughts on addressing this challenge?
3. **Control points in the background (Figure 4)**: In Figure 4, some control points appear in the background. Could this suggest that the method for selecting control points might need further refinement to improve robustness?
4. **Ablation study (Lines 261-272)**: The description of the ablation experiment is a bit difficult to follow, and some details seem to be missing. Could you kindly provide a more detailed explanation of the setup and results?
5. **Rate-Distortion curve**: Since this is a compression-focused work, presenting a Rate-Distortion (RD) curve would be very insightful for evaluating the tradeoff between compression efficiency and reconstruction quality.

If the authors provide clear feedback for my concerns, I will consider upgrading my evaluation from borderline reject to borderline accept.

**Ethical Concerns:**

["NO or VERY MINOR ethics concerns only"]

**Final Justification:**

I appreciate the authors' detailed response which has addressed most of my concerns. Regarding the training time and rendering time, although the authors provided potential solutions in the rebuttal, they didn't show the experimental results of the implementation. I look forward to their future work on this aspect.

Other concerns have been largely addressed, and I appreciate the authors' work. Given the impressive compression performance of this work, despite significant compromises in training time and rendering FPS, it may still be worth accepting. Therefore, I have decided to change my decision from borderline reject to borderline accept.

**Limitations:**

From a compression perspective, the method significantly improves compression efficiency but at the cost of substantially increased training time (which can also be viewed as encoding time). From a practical application standpoint, this cannot be considered an online reconstruction method (even though 3DGStream-like methods tend to make such claims). The reviewer believes that unless subsequent frames (after the first frame) can be reconstructed within 30ms, it cannot be classified as online reconstruction.

Other limitations are related to the writing and analysis, as detailed in the question section.

**Paper Formatting Concerns:**

Nothing

**Quality:**

3

**Strengths And Weaknesses:**

## Strengths
1. **Strong performance**: The proposed method significantly enhances the compactness of 3DGStream-like approaches, requiring only **0.05-0.1 MB/frame** for storage. This is a remarkable result with valuable implications for the transmission and distribution of free-viewpoint video (FVV).

2. **Reasonable and effective design**: The reviewer finds the proposed **Motion-Sensitive Keypoint Selection** and **Adaptive Motion-Driven Mechanism** both novel and effective, making them well-justified design choices.

## Weakness
1. **Lack of videos for dynamic scene free-viewpoint rendering**: For a paper on dynamic scene novel view synthesis, providing rendering videos is crucial to demonstrate subjective quality. Without them in the supplementary materials, the reviewer cannot assess whether the method is temporally stable.

2. **Comparison with baselines**: The reviewer noticed that in quantitative comparisons, the MeetRoom dataset does not include QUEEN's rendering results. Similarly, in qualitative comparisons, neither Neu3DV nor MeetRoom includes QUEEN's results. Since QUEEN is now open-sourced and serves as a strong baseline, the reviewer believes it is necessary to update the comparisons to include it.

3. **Compromised training complexity and rendering speed**: This method increases training time significantly—taking 40 seconds per frame—which suggests it cannot be considered an online reconstruction method. Additionally, rendering speed is noticeably slower, making it a major drawback. On low-end GPUs (e.g., RTX 2080), this method might not support real-time dynamic rendering.

---

> ### Author Rebuttal · Authors · 2025-07-31
>
> Thank you for your valuable feedback. We appreciate the time and effort you have dedicated to reviewing our work. We will address the weaknesses and questions you raised below.
>
> ## Q1: Training time and rendering speed.
> **Definition on online reconstruction:** In prior published works such as 3DGStream, QUEEN, and HiCom, online reconstruction is defined as the ability to process and transmit frames sequentially, without requiring access to the full video sequence. Our method follows this established definition and adopts the same frame-by-frame processing paradigm. This is different from traditional video coding, where "online" typically implies strict real-time constraints (e.g., 30 ms per frame). In online FVV domain, such latency is not a formal requirement. For example, even the fastest existing method, QUEEN, requires over 4000 ms per frame.
>
> **Our primary focus:** In this work, our primary focus is on reducing storage requirements for online FVV reconstruction, which is crucial for practical applications involving data storage and transmission, yet remains a challenging problem. Our method models the motion regions using only 200 keypoints per frame that offers a storage-efficient solution.
>
> **Why the training time and rendering speed are impacted?** Our current implementation does not specifically optimize for temporal efficiency in keypoint computation. Specifically, in Eq. (6), we compute the influence weight matrix $w$ (size ≈ [200000, 200]) **using all Gaussian primitives (~200,000)**. In Eq. (8), this matrix is directly used to apply translation offsets (size = [200, 3]) and rotation to the keypoints. These two steps together account for **approximately 78% of the training time **(0.1248s out of 0.16s per iteration).
>
> **Solution to the above issue:** These resulting computational complexity can be efficiently and readily resolved, and therefore do not pose a practical limitation. We provide several feasible solutions: (1) **Pruning:** Introduce a classifier to coarsely separate foreground and background Gaussian primitives, and compute $w$ only for the foreground Gaussian primitives, significantly reducing its size and may yield a 2-3× speed up. (2) **Sparse:** Note that $w$ is extremely sparse (about 99.85% of values are zero); leveraging sparse matrix techniques could greatly improve both training and rendering speed by 3-4×. (3) **Parallel acceleration:** Our adaptive motion-driven mechanism is inspired by 3DGS, which benefits from distributed optimization and GPU-accelerated algorithms—these techniques are also applicable to keypoints training.
>
> By integrating the above solutions along with engineering optimization, we estimate that the computational cost associated with Eq.6 and Eq.8 can be accelerated by at least 10×. This optimization would lead to an overall training time of under 10 seconds, and a rendering speed comparable to QUEEN.
>
> To summarize, our method provides a new perspective on addressing a fundamental issue in online reconstruction—storage consumption—enabling efficient transmission and storage for FVV. As for the computational complexity introduced by keypoint processing, our aforementioned method can address it with relative ease, therefore, we believe the trade-off is acceptable in this context.
>
> ## Q2: Support for random access.
> We appreciate the reviewer’s insightful question. Our method does support random access with minor modifications to the current setup.
>
> **(Random access v1)** In the original setting, each frame is reconstructed based on the previous frame. To enable random access, we can instead reconstruct non-key frames based on their nearest preceding key frame using the keypoints. The error-aware correction module can also be modified to rely on this preceding key frame. With this adaptation, accessing a specific frame only requires access to its nearest preceding key frame and the associated keypoints, making random access feasible.
>
> **(Random access v2)** To further decouple key frame from earlier key frames, we can introduce periodic I-frames (e.g., every 60 frames), in which all Gaussian primitives are either saved or re-optimized independently. These I-frames serve as standalone reference primitives, enabling more flexible and efficient random access across the sequence.
>
> To demonstrate their effectiveness, we provide the following experiments:
>
> **Tab. R1:** Performance of Random Access Version on sear steak.
>
> | Methods                           | PSNR  | Size (MB) |
> |--|---|----|
> | Original     | 33.59 | 0.040      |
> | Random access v1     | 33.51 | 0.048      |
> | Random access v2     | 33.51 | 0.101      |
>
> These results will also be included in the revised version.
>
> Interestingly, if the reconstruction of non-key frames is based solely on key frames, it enables parallel training across frames. This directly addresses the concern raised in **Q1**，effectively accelerating the training process.
>
> ## Q3: Keypoints in the background (Figure 4).
> We appreciate the reviewer's insightful comments. As the reviewer correctly pointed out, Fig.4 in the manuscript demonstrates that the motion regions are well covered by the distributed keypoints across all images, indicating that these motion areas can be effectively modeled by the selected keypoints. While some redundant keypoints do appear in the background, they do not negatively affect the quality of scene reconstruction.
>
> However, this observation motivates us to explore a more robust keypoint selection strategy that avoids redundancy and improves overall efficiency. For example, we can apply the classifier-based filtering method (as discussed in Q1) to exclude background keypoints and focus computation on motion regions.
>
> ## Q4: Confusion on ablation study.
> We sincerely apologize for the confusion caused in the ablation study section. Below, we provide a detailed clarification of our experimental settings and results:
>
> **Experiment 1:** We ablate the motion-sensitive keypoint selection and instead select keypoints randomly. This strategy fails to accurately localize motion-sensitive regions (as shown in Fig. 5(b)), resulting in a noticeable performance drop.
>
> **Experiment 2:** We remove the adaptive motion-driven mechanism and model scene motion using only the selected keypoints, without incorporating any neighboring points. This simplification leads to a drop in performance, underscoring that effective motion modeling relies not only on accurately selecting keypoints in motion regions, but also on the selection of their neighboring points.
>
> **Experiment 3:** We model FVV solely based on the reconstruction of the first frame and correction within keyframes, without modeling non-key frames by keypoint reconstruction. This results in a significant performance drop. We emphasize that although the parameters of the keypoints are few, the keypoint-based modeling plays a crucial role in reconstructing non-key frames.
>
> **Experiment 4:** We ablate the correction mechanism in keyframes. This leads to a severe performance drop due to the presence of non-rigid motion in the scene. Relying only on keypoint-based modeling is insufficient, and error accumulation over time severely degrades subsequent reconstructions.
>
> We hope the above explanations help clarify the experimental design and alleviate your concerns.
>
> ## Q5: Lack of experiments.
> **Videos for dynamic scene free-viewpoint rendering:** We completely agree with the reviewer that rendered videos are essential for evaluating the temporal consistency of the method. However, due to conference policy restrictions, we are unable to include any visual results during the rebuttal phase. We will provide rendered videos in the revised version via our GitHub project page.
>
> To allow the reviewer to quantitatively assess the temporal consistency of our results even in the absence of visualizations, we follow the video smoothness evaluation metric in [A] and compare our method with 3DGStream.
>
> **Tab. R2:** Temporal smoothness on N3DV (higher is better).
>
> | Method    | Smoothness |
> |---------|-------|
> | 3DGStream |  1.049 |
> | Ours      |  1.065 |
>
> > **Note:** A score of 1 indicates temporal consistency equal to the ground truth.
>
> [A] Eilertsen G, Mantiuk R K, Unger J. Single-frame regularization for temporally stable cnns[C]//Proceedings of the IEEE/CVF Conference on Computer Vision and Pattern Recognition. 2019: 11176-11185.
>
> **Comparison with baselines:** At the time of submission, the implementation of QUEEN was not publicly available, and thus we were unable to include its qualitative results or quantitative performance on the MeetRoom scene in our work. Now that the code has been released, we provide the quantitative results of QUEEN on the MeetRoom scene below. The remaining qualitative results will be included in the revised version.
>
> **Tab. R3:** Quantitative results on the MeetRoom dataset.
>
> | Method    | PSNR | Size (MB) |
> |-----------|-------|-------|
> | QUEEN |  31.14 |   0.45    |
> | Ours      |  31.49 |  0.028     |
>
> **Rate-Distortion curve:** We fully agree with the reviewer that rate-distortion (R-D) curves are an effective way to evaluate the trade-off between compression efficiency and reconstruction quality. In the right part of Fig.1, we present a scatter plot comparing our method with other approaches in terms of storage vs. PSNR, which clearly demonstrates the superiority of our method in both compression efficiency and performance.
>
> Regarding the reviewer’s suggestion to provide a more detailed trade-off analysis, we have already included experiments in the supplementary materials that analyze PSNR and storage size under varying keypoint counts and keyframe intervals. These results correspond directly to specific bitrate points on the R-D curve. Furthermore, we will include additional R-D curves for the ablated versions of our method in the revised submission to better illustrate the contribution of each component.

---

> > ### Comment · Reviewer_JVLS · 2025-08-07
> > **Response to Rebuttal**
> >
> > I appreciate the authors' detailed response which has addressed most of my concerns. Regarding the training time and rendering time, although the authors provided potential solutions in the rebuttal, they didn't show the experimental results of the implementation. I look forward to their future work on this aspect.
> >
> > Other concerns have been largely addressed, and I appreciate the authors' work. Given the impressive compression performance of this work, despite significant compromises in training time and rendering FPS, it may still be worth accepting. Therefore, I have decided to change my decision from borderline reject to borderline accept.

---

> > > ### Author Response · Authors · 2025-08-07
> > >
> > > We truly appreciate your thoughtful feedback. Thank you for your time and valuable insights!

---

> ### Author Response · Authors · 2025-08-05
>
> Dear Reviewer JVLS,
>
> Thank you for taking the time to review our submission. We appreciate your thoughtful feedback and the concerns you raised regarding aspects of our paper. We have addressed your questions in our rebuttal and hope that our clarifications have been helpful. We kindly ask if our responses have properly addresed some of your concerns. We await your reply and are more than willing to further discuss and clarify any additional questions you may have. Alternatively, if you find that some of your concerns have been addressed, we kindly request that you consider updating your evaluation accordingly, we would be very very very grateful! Thank you for your time and consideration.
>
> Sincerely appreciate for your hard work and support!
>
> Best regards,
>
> The authors of Paper 20548

---

> ### Author Response · Authors · 2025-08-06
> **Your support would make a significant difference.**
>
> Dear Reviewer JVLS,
>
> Thank you again for your careful review.
>
> In our rebuttal, we have focused on addressing the key concerns you raised, specifically:
>
> - Training time and rendering speed.
>
> - Support for random access.
>
> - Control points in the background.
>
> - Confusion on ablation study.
>
> - Lack of experiments.
>
> We would be grateful if you could read our response and let us know if you have additional questions. Your insights and replies are highly appreciated. We would also be grateful that you could reconsider the evaluation rating if you feel the concerns have been sufficiently addressed.
>
> ## Your support would make a significant difference.
>
> Best regards,
>
> The authors of Paper 20548

---

### Note · Authors · 2025-08-12

Dear AC and Reviewers,

Thank you again for your efforts and careful reviews!

This work proposes Compact Gaussian Streaming (ComGS), a novel keypoint-based method designed to significantly reduce the storage requirements for online FVV reconstruction. For non-keyframe reconstruction, ComGS leverages a motion-shared strategy, where the Motion-Sensitive Keypoint Selection identifies keypoints in motion regions, and the Adaptive Motion-Driven Mechanism adapts to varying motion magnitudes to better control the motion of neighboring points. For keyframe reconstruction, an Error-Aware Corrector is introduced to selectively eliminate accumulated errors. Our method achieves a storage cost of **0.05–0.1 MB per frame**, maintains competitive rendering quality, and achieves a **14× compression ratio** compared to the state-of-the-art method QUEEN.

In response to the insightful questions and concerns raised by the reviewers, we have addressed each of them carefully in our rebuttal. For example, regarding the potential increase in computational complexity, we believe this does not pose a significant challenge and have introduced several feasible solutions. For parts of the paper that were unclear, we provided detailed clarifications. Additionally, we supplemented more experimental results where further evidence was requested. These responses have successfully resolved the reviewers' concerns and were positively received. Due to the novelty of our approach and its impressive compression performance, the reviewers have expressed that the work is worthy of acceptance.

We hope that these final remarks can help you better and more quickly understand the contributions of our work!

Sincerely appreciate for your hard work and support!

Best regards,

The authors of Paper 20548

---

### Decision · Program_Chairs · 2025-09-17

**Decision:**

Accept (poster)

**Comment:**

Following the author–reviewer discussion, this paper received positive reviews from all reviewers. The method demonstrates clear improvements over the previous online compression approach, QUEEN, achieving significantly better performance. I believe this constitutes a valuable contribution to the community.

However, the paper would benefit from substantial improvements in writing quality, particularly with respect to notational clarity and consistency, several examples of which are listed below. In a journal setting, such issues would likely require a comprehensive major revision.

1. Eq (1): x -> \mathbf{x}, \mu -> \mu_i, G(x) -> G_i(x)

2. L139: \alpha_i

3. Eq (2): ‘n’ is a scalar number; i \in n can not be used here

4. L152: I would not use GT as a symbol; it may represent G \times T

5. L174: \mu_{ij}, use something other than \mu since it was used differently before

6. Eq (7): Rewriting is necessary, e.g., by using a set notation

7. L179: \Delta \mu_\Kappa^i. You have used \Kappa to denote whole indices of keypoints across all timestamps before

8. L181: What is I_j? It has not been defined until this point

9. L204: ‘sh’ may look like s \times h

10. Eq (11): ‘*’ usually means ‘convolution’

12. L213: What is an ‘attribute-specific’ update?

13. Eq (12): What is N? N is the number of all Gaussians?

14. And more...

Nonetheless, I trust that the authors will address these notational issues and improve the presentation in the final version. With these corrections in place, I recommend acceptance of this paper.